# *Serendipita indica* E5′NT modulates extracellular nucleotide levels in the plant apoplast and affects fungal colonization

Shadab Nizam[1,2], Xiaoyu Qiang[1,2], Stephan Wawra[2], Robin Nostadt[1], Felix Getzke[2], Florian Schwanke[2], Ingo Dreyer[3] (iD), Gregor Langen[2] (iD) & Alga Zuccaro[1,2,*] (iD)

## Abstract

Extracellular adenosine 5′-triphosphate (eATP) is an essential signaling molecule that mediates different cellular processes through its interaction with membrane-associated receptor proteins in animals and plants. eATP regulates plant growth, development, and responses to biotic and abiotic stresses. Its accumulation in the apoplast induces ROS production and cytoplasmic calcium increase mediating a defense response to invading microbes. We show here that perception of extracellular nucleotides, such as eATP, is important in plant–fungus interactions and that during colonization by the beneficial root endophyte *Serendipita indica* eATP accumulates in the apoplast at early symbiotic stages. Using liquid chromatography–tandem mass spectrometry, and cytological and functional analysis, we show that *S. indica* secrets *Si*E5′NT, an enzymatically active ecto-5′-nucleotidase capable of hydrolyzing nucleotides in the apoplast. *Arabidopsis thaliana* lines producing extracellular *Si*E5′NT are significantly better colonized, have reduced eATP levels, and altered responses to biotic stresses, indicating that *Si*E5′NT functions as a compatibility factor. Our data suggest that extracellular bioactive nucleotides and their perception play an important role in fungus–root interactions and that fungal-derived enzymes can modify apoplastic metabolites to promote fungal accommodation.

**Keywords** DORN1; extracellular nucleotides; lectin receptor kinase LecRK-I.9; *Piriformospora indica*; purinergic signaling

**Subject Categories** Microbiology, Virology & Host Pathogen Interaction; Plant Biology

## Introduction

ATP is a coenzyme that serves as a universal energy currency and as a building block of nucleic acids and secondary metabolites.

Additionally, ATP and adenosine are important extracellular regulators in plants and animals [1–5]. In plants, it has been shown that cells release ATP during growth, physical injury, and in response to various abiotic and biotic stresses and microbe-derived elicitors [4–13]. Accumulation of extracellular ATP (eATP) triggers the production of reactive oxygen species (ROS), nitric oxide, callose deposition, cytoplasmic calcium increase, transient phosphorylation of MPK3 and MPK6, and expression of genes involved in plant stress response and immunity [8,14]. To be physiologically relevant, eATP must be sensed by specific receptors at the cell surface. Recently, the *Arabidopsis thaliana* lectin receptor kinase DORN1 (DOes not Respond to Nucleotides 1, At5g60300), also known as the LecRK-I.9 (lectin receptor kinase I.9), was identified as the first plant purinoceptor essential for the plant response to eATP, unveiling an eATP signaling pathway in plants [8]. The RXLR-dEER effector protein IPI-O secreted by the oomycete pathogen *Phytophthora infestans* targets LecRK-I.9/DORN1 [15–18], suggesting that some microbes have the tools to manipulate host eATP perception/signaling. LecRK-I.9/DORN1 mutant plants show enhanced susceptibility to pathogen infections such as the oomycete *P. brassicae* and the bacterium *Pseudomonas syringae* (*Pst*). Consistently, overexpression of LecRK-I.9/DORN1 increases plant resistance to *Phytophthora* spp. and *Pst* [19]. Taken together, these results suggest a role for an eATP-receptor protein in plant immunity.

There is increasing evidence that fungi, both pathogenic and mutualistic, have large repertoires of secreted effectors. Some were functionally shown to be able to manipulate the host cell physiology, suppress plant defense, and ultimately promote fungal colonization [20]. Among fungi, compatibility factors are mainly reported from biotrophic and hemibiotrophic leaf pathogens [21]. In contrast, only few factors have been functionally characterized from root mutualistic fungi [22–25]. Therefore, the basis and mode of action of mutualistic fungal compatibility factors remain poorly understood.

The filamentous root endophyte *Serendipita indica* (syn. *Piriformospora indica*) belongs to the order *Sebacinales* (Basidiomycota). *Serendipita indica* colonizes the root epidermal and cortex cells

---

1 Max Planck Institute for Terrestrial Microbiology, Marburg, Germany
2 Botanical Institute, Cluster of Excellence on Plant Sciences (CEPLAS), Cologne Biocenter, University of Cologne, Cologne, Germany
3 Centro de Bioinformática y Simulación Molecular (CBSM), Universidad de Talca, Talca, Chile
 *Corresponding author. Tel: +49 2214 707170; E-mail: azuccaro@uni-koeln.de

without penetrating the central cylinder and displays a biphasic colonization strategy [26–29]. During the initial phase of biotrophic colonization, the fungus invades the root cells inter- and intracellularly. Subsequently, *S. indica* switches to a host cell death-associated phase, although a defined switch to necrotrophy with massive cell death does not occur [26,27,29]. *Serendipita indica* colonization exhibits various effects on host plants including enhanced growth, improved assimilation of nitrate and phosphate, increased tolerance to abiotic stresses, and resistance against pathogens [30–33]. Since *S. indica* establishes symbiotic interactions with a wide range of experimental hosts, including the dicot model plant *A. thaliana* and the monocot cereal crop *Hordeum vulgare* (barley), it represents an excellent model system to study the role of extracellular bioactive nucleotides and eATP-mediated plant responses in the roots of unrelated hosts.

In order to identify *S. indica* secreted effectors, proteins present in the apoplastic fluid (APF) of colonized barley roots were analyzed at three different symbiotic stages by liquid chromatography–tandem mass spectrometry (LC-MS/MS). One secreted fungal protein consistently found in the apoplast at all time points is a predicted 5′-nucleotidase. The gene encoding this enzyme is induced during colonization of both barley and *A. thaliana* but not in axenic culture. Animal ecto-5′-nucleotidases (E5′NTs) have been considered to play a key role in the conversion of AMP to adenosine, counteracting eATP release from stimulated cells and further purinergic signaling together with ecto-nucleotide pyrophosphatase/phosphodiesterase (E-NPP), ecto-nucleoside triphosphate diphosphohydrolase (E-NTPDase), and alkaline phosphatases (AP) [34,35]. The importance of bioactive nucleotide-triggered signaling and fungal extracellular E5′NT activity during plant–fungus interactions is unknown. We show here that *S. indica* E5′NT is capable of hydrolyzing ATP, ADP, and AMP to adenosine and phosphate, altering eATP levels and plant responses to fungal colonization. Considering the important role E5′NT plays in *S. indica* accommodation at early symbiotic stages, we propose that modulation of extracellular nucleotide levels plays a role in compatibility during plant–fungus interactions.

## Results

### Identification of fungal proteins in the apoplast of barley roots

In order to identify soluble secreted candidate effector proteins during *S. indica* root colonization, the proteins present in the APF of barley roots at three different symbiotic stages, 5, 10, and 14 days postinoculation (representing the biotrophic, early, and late cell death-associated phases) were analyzed together with the proteins found in the culture filtrate (CF) obtained from *S. indica* axenically grown in liquid complex medium (CM). To assess possible

cytoplasmic contaminations, a strain constitutively expressing an *S. indica* codon-optimized GFP under the *Si*GPD promoter [36] was used. The absence of fungal cytoplasmic GFP protein contamination due to cell lysis was corroborated by Western blotting of the apoplastic protein samples using anti-GFP antibody (Appendix Fig S1).

LC-MS/MS analysis after tryptic digestion led to the identification of 102 *S. indica* proteins putatively targeted to the apoplast of which 33 were present at all three time points in at least one of the biological replicates (Table EV1, Dataset EV1 and Fig 1A and B). Predictions using ApoplastP (http://apoplastp.csiro.au/) and SecretomeP (http://www.cbs.dtu.dk/services/SecretomeP-1.0/) indicated that 48 of the 102 *S. indica* proteins are putatively targeted to the plant apoplast (47%) with nine proteins predicted to be secreted via a non-canonical secretion pathway (Table EV2). No peptides for GFP were found in any of the apoplastic fluid samples, confirming the GFP Western blotting analysis. Twenty proteins were uniquely identified at 5 dpi, 4 at 10 dpi, and 21 at 14 dpi, suggesting differential secretion of *S. indica* proteins at different stages of colonization (Table EV1).

Twenty-one of the 102 identified apoplastic proteins were also found in the culture filtrate of *S. indica* grown in CM, which included a large proportion of enzymes acting on peptide bonds (Fig 1A and B). The majority of these proteases might conceivably play a role in nutrition rather than in suppression of host defense because of the large overlap with the culture filtrate proteome and because amino acid uptake seems to be an active process during root colonization based on transcriptomic data [27,28,37]. Four peptidases (PIIN_00628, PIIN_02239, PIIN_02952, and PIIN_07002), however, were specifically present in the apoplastic fluid, which makes them good candidate effectors possibly involved in host defense suppression (Table EV1).

Remarkably, 19 of the 33 common proteins found in the APF at all three time points were not present in the CF samples, implying an induced secretion for these proteins during interaction with the plant roots (Table EV1, and Fig 1B). We did not identify any peptides derived from the *S. indica* bacterial endosymbiont *Rhizobium radiobacter* (syn. *Agrobacterium tumefaciens*, syn. *Agrobacterium fabrum*) [38,39] in the APF or CF samples, suggesting that *R. radiobacter*'s contribution to successful symbiosis does not involve accumulation of bacterial proteins in the apoplast of barley.

In a first step for functional interpretation of the resultant fungal protein list from the APF and CF, we assigned Gene Ontology terms [40–42]. GO term enrichment analysis for biological processes showed an over-representation of GO terms associated with proteolysis in the culture filtrate obtained from axenically grown *S. indica*, whereas the apoplastic fungal proteins were related to purine metabolism and specifically to ATP metabolic processes (Table EV3, Appendix Figs S2 and S3).

Among the fungal proteins specifically present in the apoplastic fluid of colonized roots, several ATP-scavenging enzymes, including

**Figure 1. Identification of *S. indica* apoplastic proteins.**

A   Distribution of *S. indica* apoplastic proteins identified by LC-MS/MS analysis from different symbiotic stages (5, 10, and 14 dpi) and in relation to proteins identified in CF of *S. indica* axenically grown in CM. In total, 102 *S. indica* proteins were identified in the APF of colonized barley roots. Of these, 33 proteins were present at all three time points. Twenty proteins were unique at 5 days postinfection (dpi), 4 at 10 dpi, and 21 at 14 dpi.

B   Heat map showing absolute counts of unique peptides for *S. indica* apoplastic proteins present at all three time points and in culture filtrate. S1/S2/S3 = biologically independent samples 1, 2, and 3; d = deglycosylated.

**A**

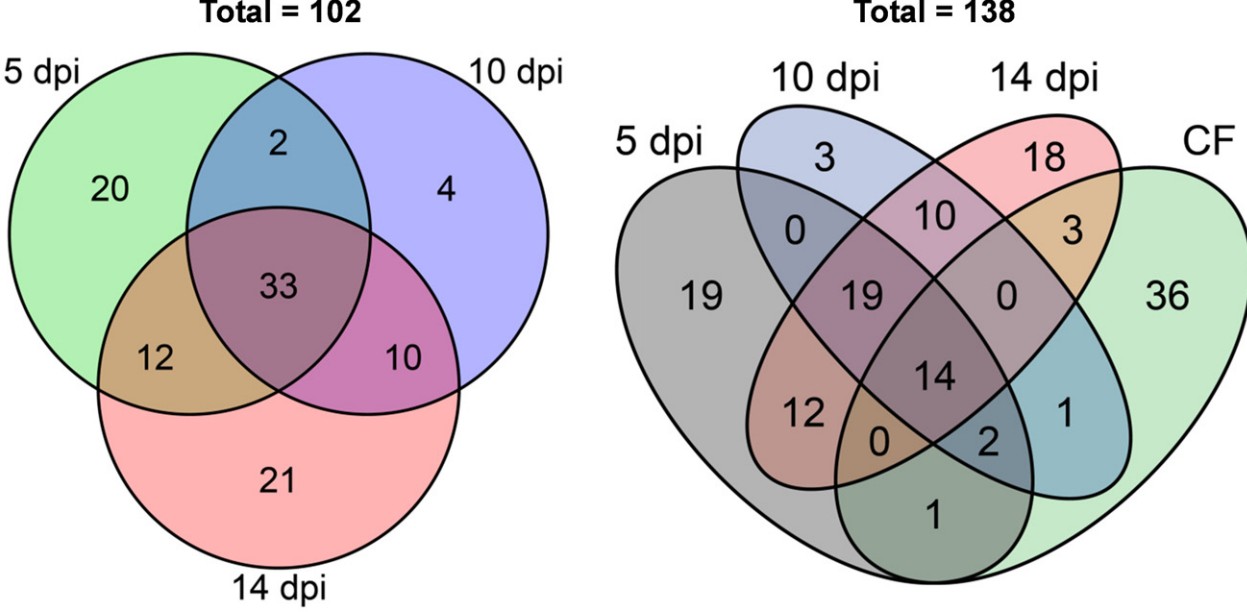

**B**

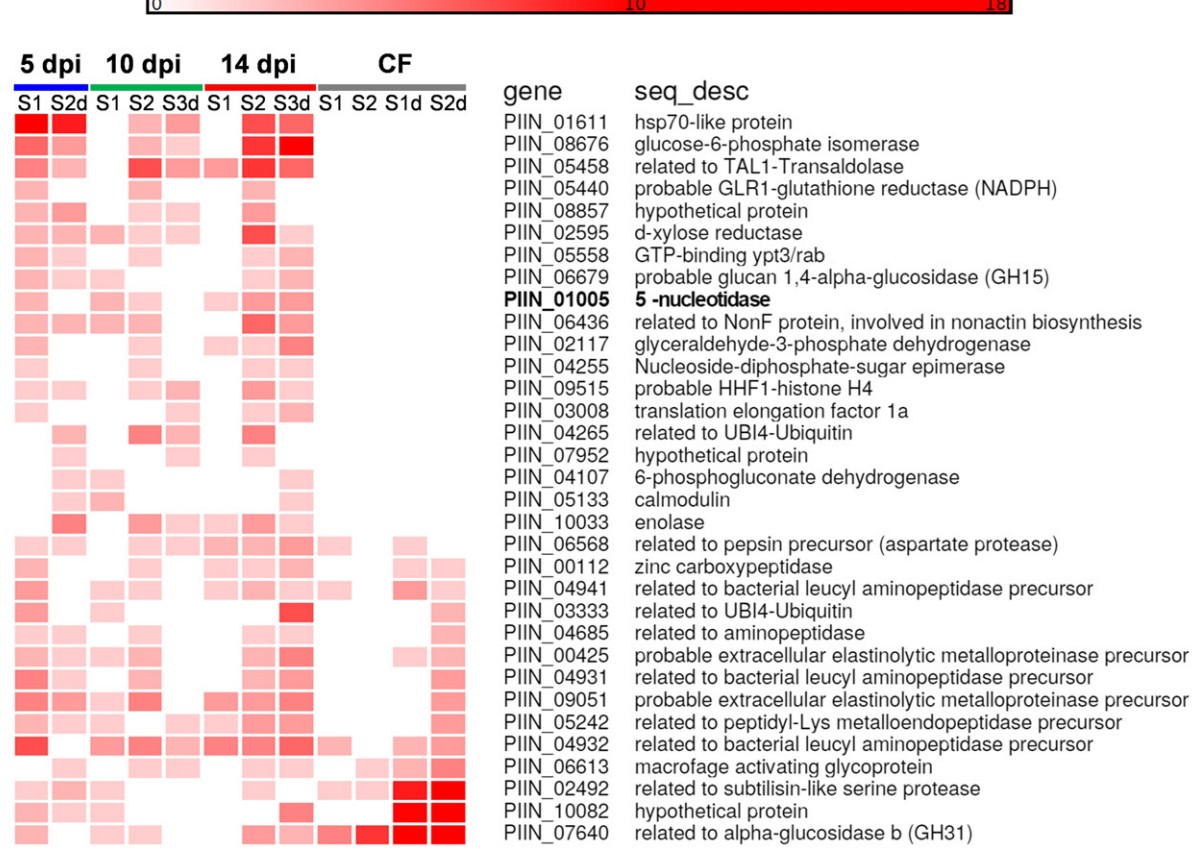

**Figure 1.**

a 5′-nucleotidase (PIIN_01005) and a nucleoside diphosphate kinase (PIIN_00784), were identified (Table EV1 and Appendix Figs S2 and S4). Twenty-one orthologues of the *S. indica* apoplastic proteins were also found in the APF of rice leaves infected by the rice blast pathogen *Magnaporthe oryzae*, including an orthologue of the secreted *S. indica* nucleoside diphosphate kinase (Table EV4 and Appendix Fig S2 asterisks) [43]. The presence of fungal-derived ATP-scavenging enzymes in the apoplast of colonized plants indicates that extracellular nucleotides may be important during plant–fungus interactions.

### eATP plays a role during fungal colonization in barley and in *Arabidopsis*

The enrichment for GO terms related to ATP metabolic processes among the fungal-derived apoplastic proteins led us to test whether root colonization by *S. indica* triggers changes in eATP levels at different symbiotic stages and in different plant species. We used a specific luminescent ATP detection assay to assess eATP levels of *Arabidopsis* seedlings grown in liquid medium or barley root apoplastic fluid samples both from *S. indica*-colonized and from mock-treated plants at 3, 5, 7, and 10 dpi. It was not possible to extract apoplastic fluid from the roots of *Arabidopsis* seedlings for these experiments because this young tissue is very delicate and fragile. All our efforts to collect apoplastic fluids ended up with cytoplasmic contaminations (measured by the LC-MS/MS proteomic approach) in this host. The results show that eATP levels are significantly higher in *S. indica*-colonized roots compared to mock-treated plants, especially at early symbiotic stages where the biotrophic interaction is still predominant (Fig 2A and B). Thus, there is release/accumulation of eATP during *S. indica* biotrophic colonization of both *Arabidopsis* and barley.

To investigate whether eATP detection plays a role during root colonization, we used an eATP-insensitive *A. thaliana* mutant, *dorn1-3* defective in the receptor LecRK-I.9 [8]. After confirming the inability of the *dorn1-3* mutant to respond to eATP and eADP with cytoplasmic calcium influx (Appendix Fig S5), colonization of this line along with the appropriate aequorin-expressing Col-0 control (Col-0$^{aq}$) was assessed by qPCR at different time points (3, 5, 7, and 10 dpi). We observed that the *dorn1-3* mutant supports significantly more fungal colonization in comparison with Col-0$^{aq}$ control, especially at early symbiotic stages (Fig 2C). Importantly, these observations strongly indicate that detection of eATP by the host affects root colonization.

### *Serendipita indica* PIIN_01005 is an ecto-5′-nucleotidase capable of hydrolyzing ATP, ADP, and AMP

Since our data show that detection of eATP plays a role during root–*S. indica* interaction, we hypothesized that this fungus has evolved an intrinsic mechanism to manipulate extracellular nucleotides levels to counteract eATP-mediated host immune responses. The secreted *S. indica* PIIN_01005, consistently found in the apoplastic fluid but not in the culture filtrate samples, is a suitable candidate for such a mechanism because of its putative hydrolytic activity on nucleotides. Microarray analysis of fungal transcripts [27,28,37] and gene expression experiments on colonized *H. vulgare* and *A. thaliana* root tissue showed that PIIN_01005 is induced during

root colonization in both experimental hosts but not in CM (Table EV1 and Fig 2D), confirming our results of the LC-MS/MS analysis. Moreover, the expression of this gene can be induced by the addition of the nucleotides AMP and ATP to the medium (Appendix Fig S6).

Ecto-5′-nucleotidases have been found in bacteria, animals, and fungi and display significant differences in the range of substrates hydrolyzed and localization. Studies of mammalian Ecto-5′-nucleotidases have shown that a membrane-bound form and soluble forms exist. A C-terminal glycosylphosphatidylinositol (GPI) anchor for PIIN_01005 was predicted by the program FragAnchor [44]. The presence of this fungal protein in the apoplastic fluid of barley indicates that a soluble form might originate during interaction *in planta* from cleavage of the anchor by phosphatidylinositol-specific phospholipase or by proteolytic cleavage as suggested for some animal 5′-nucleotidases [45–47]. Most fungi have at least one copy of a cytosolic 5′NT without predicted signal peptide and GPI anchor, whereas the secreted ecto-enzyme members are discontinuously distributed within the fungal kingdom (Fig 3A). E5′NT members are not represented in the available genomes for the Ustilaginomycotina, Pucciniomycotina, and Saccharomycotina and only rarely found in the Eurotiomycetes. However, they are widely distributed in two fungal Ascomycota classes, the Sordariomycetes with 107 from the 121 taxa examined and the Dothideomycetes with 24 from 37 taxa. In the Basidiomycota, E5′NT members are only present in the Agaricomycotina (Fig 3A). These fungal classes are among the largest groups of fungi with a high level of ecological diversity, including many plant and insect pathogens infecting a broad range of hosts.

Sequence alignment with homologous E5′NT protein sequences from animals and bacteria showed that the loop structure formed by the amino acid residues Cys476-Pro428 in human E5′NT is absent from bacterial and *S. indica* E5′NT proteins (Fig 3B, Appendix Fig S7). It has previously been shown that this loop structure is required for dimerization of human E5′NT and that this dimeric state is important for the enzyme's narrow substrate specificity [48]. In contrast, bacterial E5′NT enzymes function as monomers and have broad substrate specificity being able to hydrolyze ATP, ADP, AMP, and other 5′-ribo and 5′-deoxyribonucleotides [49]. Our structural predictions (Fig 3B and C) suggest that the fungal E5′NT is monomeric and likely capable of hydrolyzing different nucleotides.

In order to evaluate potential substrates for *Si*E5′NT, full-length *S. indica* E5′NT was expressed in *A. thaliana* and in the phytopathogenic fungus *Ustilago maydis*, neither of which has a gene encoding a secreted E5′NT predicted in their genome. Ecto-5′-nucleotidase activity was determined by the rate of inorganic phosphate release after incubation of 50 or 100 μM of either ATP, ADP, or AMP with washed culture suspensions of *U. maydis* and with preparations of plasma membrane proteins from *A. thaliana* transgenic lines, respectively. Intact cells of the *U. maydis* control strains expressing mCherry, E5′NT without the GPI anchor, or E5′NT without the secretory signal peptide (SP) and the GPI anchor under the synthetic otef promoter were able to hydrolyze ATP and ADP to a certain extent, whereas hydrolysis of AMP occurred at a very low rate. The *U. maydis* strain expressing the full-length *S. indica* ecto-5′-nucleotidase showed significantly higher hydrolysis rates for all three nucleotides tested compared to the control strains. This

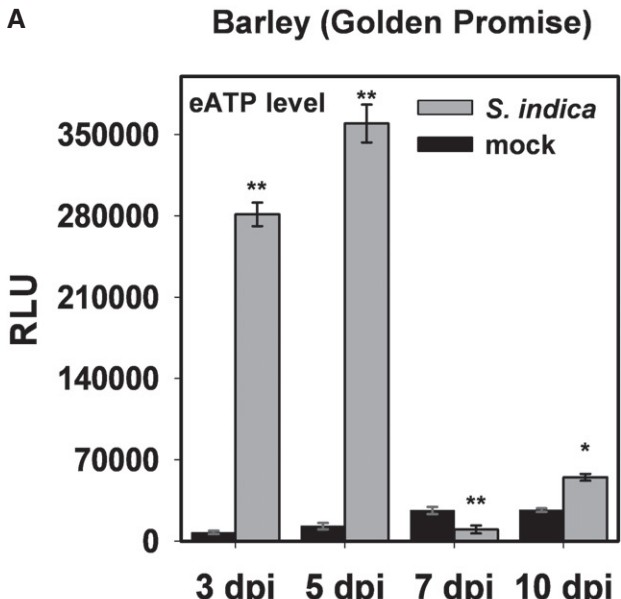

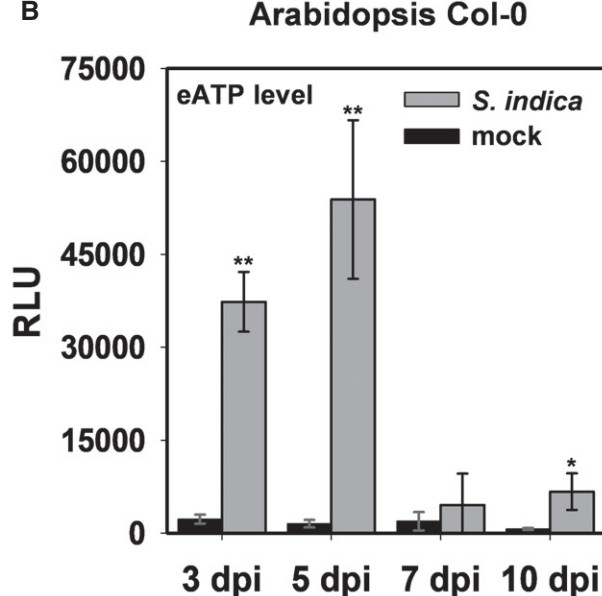

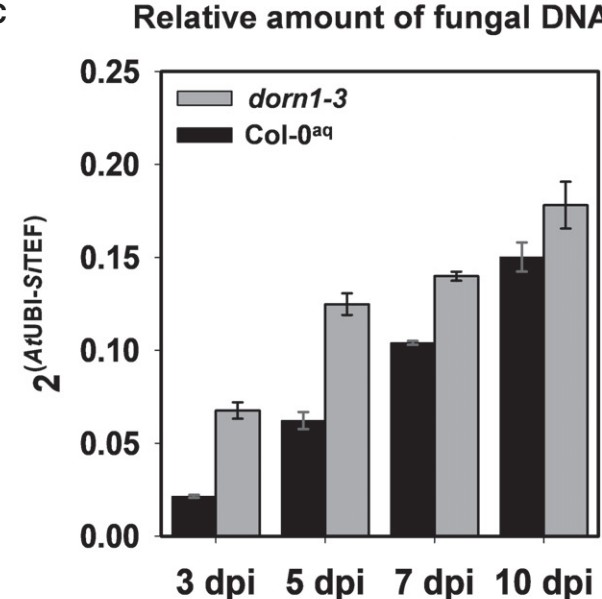

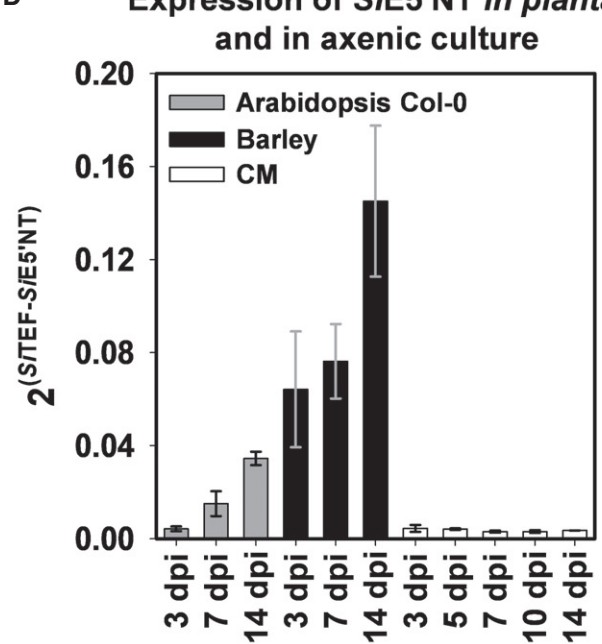

**Figure 2. *Serendipita indica* colonization leads to increased eATP levels in barley and *Arabidopsis* roots.**

A  eATP levels in APF from *S. indica*-inoculated and mock-treated barley root samples collected at the predominantly biotrophic phase (3 and 5 dpi) as well as the predominantly cell death-associated phase (7 and 10 dpi). Error bars show the standard error of the mean obtained from three independent biological replicates at $P < 0.05$ (*), $0.01$ (**) analyzed by Student's *t*-test. RLU: relative light units.

B  eATP levels measured from culture medium collected from mock-treated and *S. indica*-colonized *Arabidopsis* seedlings at 3, 5, 7, and 10 dpi. Error bars represent ±SE of the mean from three independent biological replicates. RLU: relative light units. Asterisks indicate significance at $P < 0.05$ (*), $0.01$ (**) analyzed by Student's *t*-test.

C  *S. indica* colonization of *Arabidopsis dorn1-3* mutant and the parental Col-0[aq] lines quantified by qPCR at 3, 5, 7, and 10 dpi. The ratio of fungal (*Si*TEF) to plant (*At*UBI) amplicons representing fungal colonization levels in plant root tissue was calculated using gDNA as template and the $2^{-\Delta CT}$ method. Error bars represent standard error of the mean of three technical replicates. The experiment was repeated three times for 3, 5, and 7 dpi with similar outcomes.

D  Transcript levels of *S. indica* E5′NT during colonization of barley and *Arabidopsis* at different symbiotic stages and in axenic culture. Error bars represent standard error of the mean of three independent biological replicates. CM = complex medium.

                                                                    

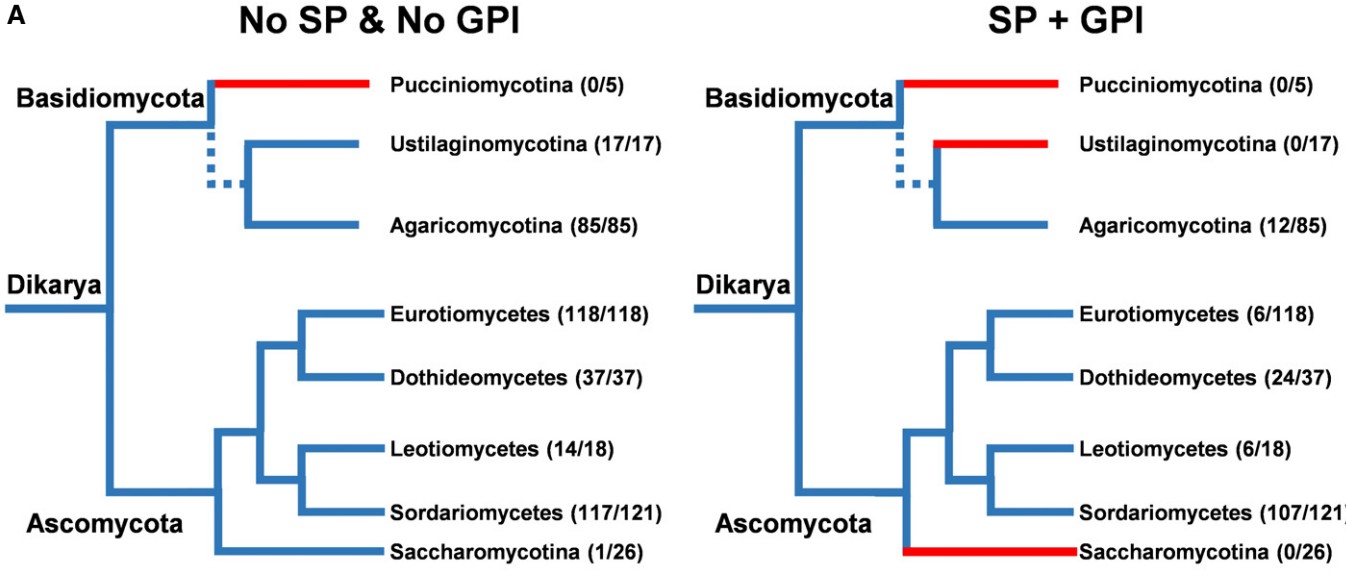

**A**

**No SP & No GPI**                                                   **SP + GPI**

**B**

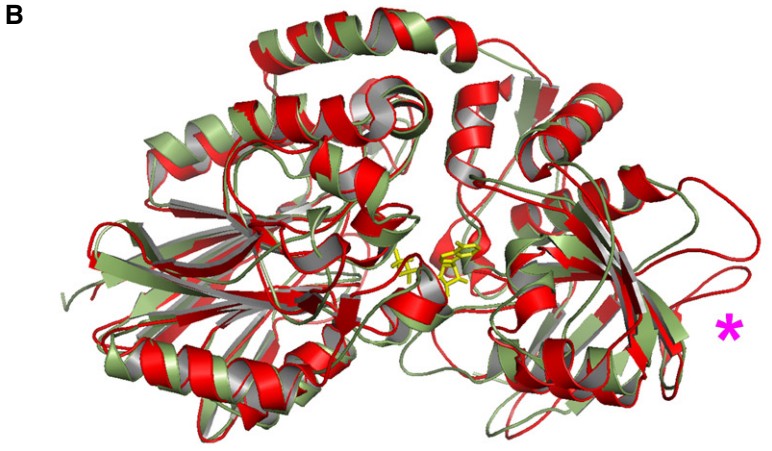

*S. indica* E5'NT &
*H. sapiens* E5'NT

**Sequence identity = 32%**

**RMSD= 2.27**
**TM-score= 0.91184**

**C**

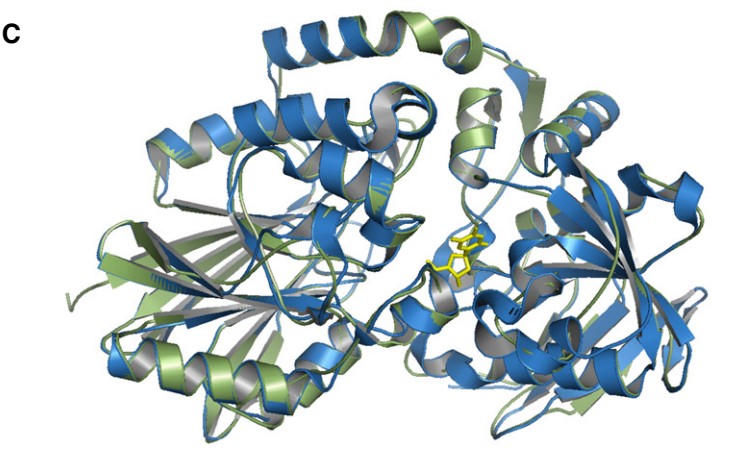

*S. indica* E5'NT &
*Thermus thermophilus* 5'NT

**Sequence identity = 36%**

**RMSD= 0.46**
**TM-score= 0.96168**

**Figure 3.**

**Figure 3. *Serendipita indica* PIIN_01005 encodes a secreted ecto-5′-nucleotidase (*Si*E5′NT).**

A Distribution of *Si*E5′NT orthologues across higher fungi. End nodes are color-coded based on the presence (blue) or absence (red) of 5′NT genes in a particular fungal taxon. Numbers in parentheses besides the nodes specify the number of species that have 5′NT genes with respect to the total number of genomes analyzed. Left tree: distribution of 5′NT without signal peptide (SP) and GPI anchor. Right tree: E5′NT with SP and GPI anchor. The distribution shows that E5′NT genes are mostly present in Ascomycota such as Sordariomycetes (107/121), followed by Dothideomycetes (24/37) and Leotiomycetes (6/18). Few species of Eurotiomycetes (6/118) possess an E5′NT orthologue. In Basidiomycota, E5′NT members are only found in the class of Agaromycotina (12/85).

B Comparison of the *Si*E5′NT structural homology model (green) with the crystal structure of human E5′NT (PDB id 4H2I, red) with 32% sequence identity. *: position of the loop involved in dimerization of human E5′NT.

C Comparison of the *Si*E5′NT structural homology model (green) with the crystal structure of *Thermus thermophiles* 5′NT (2Z1A, blue) with 36% sequence identity.

indicates that as for the monomeric bacterial variant, ATP, ADP, and AMP are potential substrates for *Si*E5′NT (Appendix Fig S8) and that *Si*E5′NT has extracellular activity. This was confirmed by the activity measurements of ectopically expressed *Si*E5′NT in *A. thaliana*. Three *A. thaliana* transgenic lines in Col-0 background expressing either full-length E5′NT (303 1L3), SP$_{E5'NT}$:mCherry:E5′NT fusion (304 6L8), or cytoplasmic mCherry (305 2L1) under the control of the 35S promoter were selected based on their expression levels in the T3 generation (Appendix Fig S9A and B). Ecto-5′-nucleotidase activity in the membrane fractions demonstrated that expression of the native form of *Si*E5′NT in *A. thaliana* leads to the production of a membrane-localized enzyme capable of hydrolyzing ATP, ADP, and AMP (Fig 4A). In contrast, the membrane fraction of the *A. thaliana* line expressing SP$_{E5'NT}$:mCherry:E5′NT fusion did not show increased activity compared to the mCherry control line, suggesting that modification at the N-terminus of the protein impacts protein activity or function. Interestingly, the absence of hydrolytic activity on AMP for *A. thaliana* lines expressing mCherry or SP$_{E5'NT}$:mCherry:E5′NT fusion implies that no E5′NT-like activity is naturally present at the membrane under this growth condition in *A. thaliana*. The lines expressing *Si*E5′NT were otherwise morphologically similar to the control lines in having the same shoot fresh weight (Appendix Fig S9C and D) but produced significantly fewer seeds during propagation (Appendix Fig S9E).

To confirm secretion of the fungal enzyme in *A. thaliana* transgenic lines, confocal microscopy was carried out with the SP$_{E5'NT}$:mCherry:E5′NT fusion line (304 6L8). A fluorescence signal in the red channel was visible at the periphery of the plant cells but not in the cytoplasm (Fig 4B, Appendix Fig S10A). *Arabidopsis* roots were treated with a 0.5 M sorbitol solution to trigger effusion of water from the cytoplasm leading to a volumetric extension of the apoplastic space and plasmolysis. After plasmolysis, the fluorescence signal in the red channel was visible at the membranes and cell walls (Appendix Fig S10B), indicating that the full-length mCherry-tagged E5′NT fusion protein is correctly secreted and that the GPI anchor is functional. However, we cannot exclude that the E5′NT part of the fusion protein is partially misfolded leading to the observed impaired hydrolytic activity in this line. No auto-fluorescence signals were detected in the UV channel and in the control Col-0 WT line (Appendix Fig S10A and B).

## SiE5′NT affects eATP levels and fungal colonization

To investigate the role of *Si*E5′NT in fungal accommodation in plant roots, the *A. thaliana* line 303 1L3 constitutively expressing the enzymatically active *Si*E5′NT (Fig 4A) was used along with the

SP$_{E5'NT}$:mCherry:E5′NT fusion line 304 6L8 and mCherry line 305 2L1 for colonization assays at 3, 5, and 7 dpi (Fig 4C and Appendix Fig S11A–C). We observed a significantly enhanced fungal colonization at all time points in the *Si*E5′NT 303 1L3 plants with respect to *A. thaliana* lines with no enhanced ecto-5′NT enzyme activities (Fig 4C). This phenotype was confirmed in independent transgenic lines (Appendix Fig S11A–C). Correspondingly, the *Si*E5′NT plants accumulated significantly lower levels of ATP in the growth medium than the control lines at 5 dpi, the time point at which we observed the highest levels of eATP accumulation upon fungal colonization (Fig 4D). Interestingly, mutation of the DORN1 receptor did not affect eATP concentrations (Fig 4D), suggesting that perception of eATP is not involved in the regulation of its extracellular abundance.

To define the transcriptional response of the *A. thaliana* transgenic *Si*E5′NT line to fungal colonization at 5 dpi, we determined the expression of AT1g58420, a marker gene reported to be induced by ATP and wounding [8]. In our experiments, AT1g58420 was highly responsive to *S. indica* colonization (Fig 4E and F). Analysis of the expression profiles of this gene upon different treatments showed that AT1g58420 is also responsive to other danger- and microbe-associated molecular patterns (DAMPs and MAMPs), including chitin and flagellin (Appendix Fig S12, and public microarrays, GENEVESTIGATOR, https://genevestigator.com/gv/). Mutation of the DORN1 receptor was shown to abolish ATP-induced gene expression and to reduce wounding-induced gene expression [8]. Upon colonization with *S. indica*, the *dorn1-3* mutant plants displayed an increased expression of the marker gene At1g58420 compared to the colonized control line Col-0$^{aq}$, which correlates to the increased fungal biomass (Fig 4E and F). This indicates that induction of this marker gene by *S. indica* is not necessarily related to increased eATP levels and its detection by DORN1. Interestingly, despite a similar level of fungal colonization, the *Si*E5′NT plants had a significantly lower induction of this marker gene compared to the *dorn1-3* mutant plants (Fig 4E and F). Expression of two additional marker genes responsive to eATP, CPK28, and RBOHD [8] was also analyzed. These genes showed a similar pattern as observed for the AT1g58420 marker gene with lower induction for the *Si*E5′NT plants compared to the *dorn1-3* mutant plants, but their induction in response to fungal colonization was not as high as for the marker gene AT1g58420 (Appendix Fig S13).

Finally, to test whether *Si*E5′NT would also increase susceptibility to fungal disease, we additionally assessed colonization of *Arabidopsis* roots by the pathogen *Colletotrichum incanum* which colonizes members of the Brassicaceae, Fabaceae, and Solanaceae and which severely inhibits *Arabidopsis* growth [50]. The *Si*E5′NT plants had a significantly higher colonization by this pathogen (Appendix Fig S14).

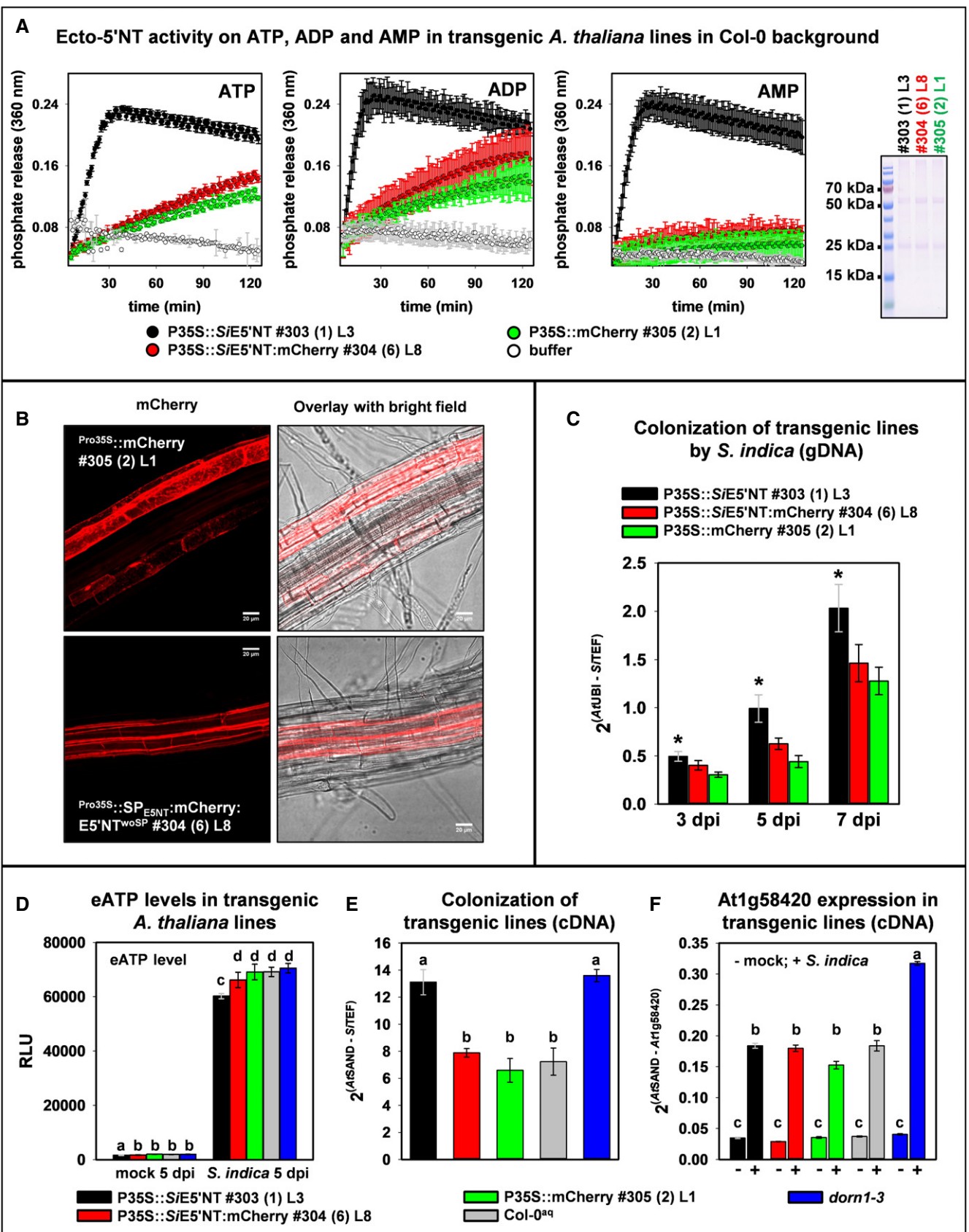

**Figure 4.**

◀

**Figure 4.  The ecto-5′-nucleotidase activity of *Si*E5′NT in the apoplast of *Arabidopsis* leads to enhanced *S. indica* colonization similar to that of the *dorn1-3* (ATP receptor) mutant line.**

A   Ecto-5′-nucleotidase activity measured in membrane protein preparations of *Arabidopsis* plants expressing ^Pro35S^::E5′NT (#303), ^Pro35S^::SP_E5′NT_:mCherry:E5′NT^woSP^ (#304), or ^Pro35S^::mCherry (#305). E5′NT activity was measured after incubation with 100 μM of either ATP, ADP, or AMP. In the membrane protein preparations from ^Pro35S^::E5′NT (#303) lines, phosphate release was specifically increased upon incubation with purines. Error bars represent the standard error of the mean from three technical repetitions. The Coomassie-stained SDS–PAGE shows the protein pattern of the membrane fractions for the individual transgenic lines. Equal volumes were loaded. The experiment was repeated two times with similar results.

B   Confocal microscopy images of *Arabidopsis* roots expressing either cytosolic mCherry (#305) or ^Pro35S^::SP_E5′NT_:mCherry:E5′NT^woSP^ (#304) showing secretion of the E5′NT fusion protein. mCherry images show z-stacks of 14 image planes of 1 μm each. Scale bar = 20 μm.

C   The transgenic *Arabidopsis* line ^Pro35S^::E5′NT (#303) expressing untagged full-length *Si*E5′NT was better colonized by *S. indica*. Error bars of the qPCR data represent ± SE of the mean from three independent biological replicates. Asterisks indicate significance (Student's *t*-test, *$P < 0.05$).

D   *S. indica* induced eATP release in different *Arabidopsis* transgenic lines. Culture medium was collected from mock-treated or *S. indica*-inoculated seedlings growing in liquid medium at 5 dpi, and released eATP was measured. RLU: relative light units. Error bars represent ± SE of the mean from three biological replicates. Letters indicate significance to all other samples within the same treatment group (ANOVA, $P < 0.05$).

E   *S. indica* colonization of transgenic lines at 5 dpi. Error bars represent ± SE of the mean from three biological replicates. Letters indicate significance to all other samples within the same treatment group (ANOVA, $P < 0.05$).

F   Expression analysis of the eATP responsive gene At1g58420 measured by qRT–PCR. Error bars represent ± SE of the mean from three independent biological replicates (independent from those shown in Fig. 2C). Letters indicate significant groups (ANOVA, $P < 0.01$, for the line 305 $P < 0.05$).

## Phosphate released from the hydrolysis of eATP could serve as a nutrient

Extracellular *Si*E5′NT activity leads to significantly decreased amounts of free eATP in the apoplast of colonized root tissue and, consequently, an increased availability of phosphate (Pi). It has previously been shown that *S. indica* transfers phosphate to the host and improves plant growth more effectively at low phosphate compared to high phosphate conditions at later colonization stages [51,52]. To evaluate the effect of an additional apoplastic Pi-source on the nutrient exchange between plant and fungus, we employed a computational cell biology approach and simulated the dynamics of a network of proton pumps and proton-coupled transporters during the early biotrophic interaction phase. We first used a model where the plant provides sugar to the fungus while the fungus provides phosphate to the plant as described for classical mycorrhizal associations [53]. When apoplastic ATP levels and subsequent Pi levels increase, the computational simulations predict that the sugar flux from the plant to the fungus is not affected but the phosphate fluxes change (Fig 5). An external Pi-source provokes an increased Pi-uptake by the plant and a reduced Pi-release or even an uptake of Pi by the fungus. Thus, the hydrolysis of eATP to adenosine and phosphate reduces the amount of Pi that the fungus has to provide to obtain sugar from the plant and could also serve for the Pi-nutrition of the fungus in the early colonization phase without affecting sugar transfer. These new insights might guide future experiments to clarify whether the activity of *Si*E5′NT in the apoplast could serve both nutritional needs and modulation of host immunity.

## Discussion

To establish an integrated and holistic view of the role of the metabolic interplay between plants and their microbes, it is of paramount importance to investigate the mechanisms by which plant-associated microbes manipulate plant-derived metabolites and how plant metabolism is linked to immunity. Apoplastic communication and its metabolic fluxes have crucial functions in mediating microbial accommodation. During root colonization, plants and microbes secrete a number of proteins to the apoplast that are important

determinants for the outcome of the interaction. Plant cells release small amounts of ATP into their extracellular matrix as they grow [7]. The eATP level can modulate the rate of cell growth in diverse tissues. Besides the physiological role in growth modulation, eATP is released to the extracellular environment in response to biotic stresses modulating defense responses, e.g., upon wounding and also when their plasma membranes are stretched during delivery of secretory vesicles [4]. This implicates a regulatory role for enzymes that can hydrolyze extracellular nucleotides and thus limit their accumulation in the extracellular environment. In plants, the level of extracellular nucleotides is controlled by various phosphatases, prominent among which are apyrases EC 3.6.1.5 (nucleoside triphosphate diphosphohydrolases, NTPDases) [4]. In animals, ecto-nucleotidases play a pivotal role in purinergic signal transmission where the major extracellular purine and pyrimidine compounds known to elicit cell surface receptor-mediated signals are ATP, ADP, UTP, UDP, UDP-glucose, some additional nucleotide sugars, some dinucleoside polyphosphates, and the nucleoside adenosine [2,54]. The four major groups of ecto-nucleotidases involved in extracellular purinergic signaling include the ecto-nucleoside triphosphate diphosphohydrolases (E-NTPDases), ecto-5′-nucleotidase (eN), ecto-nucleotide pyrophosphatase/phosphodiesterases (E-NPPs), and alkaline phosphatases (APs) [55]. Depending on subtype, ecto-nucleotidases typically hydrolyze nucleoside tri-, di-, and monophosphates and dinucleoside polyphosphates producing nucleoside diphosphates, nucleoside monophosphates, nucleosides, phosphate, and inorganic pyrophosphate (PPi) [55]. The extracellular membrane-bound ecto-5′-nucleotidase CD73 is anchored to the plasma membrane via a glycosyl phosphatidylinositol (GPI) anchor [55,56]. This enzyme hydrolyzes AMP to adenosine where ADP and ATP are competitive inhibitors, with inhibition constants in the low micromolar range [48]. The CD73 together with the ecto-apyrase CD39 that catalyzes the phosphohydrolysis of ATP and ADP to AMP is part of a cascade to terminate the action of nucleotides as extracellular signaling molecules by decreasing the levels of eATP and eADP and by generating adenosine. In animals, this latter bioactive nucleoside can activate the purinergic ligand-gated ion channel P2X and the G protein-coupled P2Y receptors, counteracting ATP-induced signaling [55]. A purinergic receptor for extracellular adenosine or other bioactive nucleosides is not known in plants,

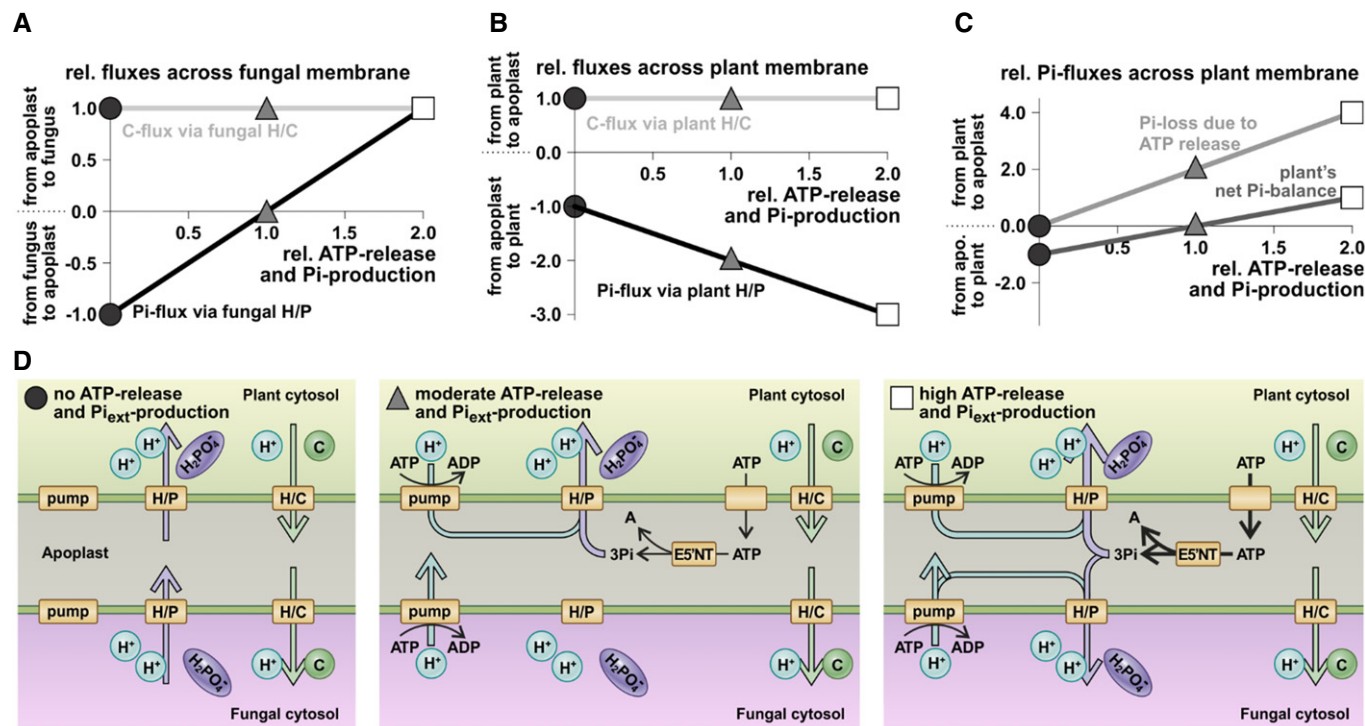

**Figure 5. Simulation of the effect of eATP release and its cleavage in the apoplast on the nutrient exchange between plant and fungus.**

A   Relative fluxes of phosphates (Pi) and sugars (C) across the fungal plasma membrane. Without additional Pi-source in the apoplast (value 0 at the *x*-axis, black dots), there is a constant flux of phosphate via the proton-coupled phosphate (H/P) transporter from the fungus to the apoplast and a constant flux of sugar from the apoplast to the fungus. For better comparison, these fluxes were normalized to 1 and −1, respectively, and all other fluxes were calculated relative to these control values. With increasing ATP-release and Pi-production, the Pi-efflux gets smaller and is zero at a relative Pi-production rate of 1 (gray triangles). At higher ATP-release and Pi-production rates, the fungus imports Pi, i.e., the transport direction of phosphate has been inverted in comparison with the control condition (white square). The C-flux via the proton-coupled sugar (H/C) transporter is not affected by the additional Pi-source (horizontal gray line).

B   Relative fluxes of Pi and C across the plant plasma membrane. Without additional Pi-source in the apoplast (value 0 at the *x*-axis, black dots), there is a constant flux of sugar via the H/C transporter from the plant to the apoplast (and thereafter to the fungus) and a constant flux of phosphate (coming from the fungus) via the H/P transporter from the apoplast to the plant. With increasing ATP-release and Pi-production, the Pi-influx via the H/P transporter increases while the C-flux via the H/C transporter is unaffected by the additional Pi-source (horizontal gray line).

C   Phosphate fluxes across the plant plasma membrane. Besides the Pi-uptake via the H/P transporter (B, black line, for clarity not shown in C), the plant loses Pi due to the ATP-release (light gray line). The difference between Pi-uptake and Pi-loss is the net Pi-balance of the plant (dark gray line). Without ATP-release (value 0 at the *x*-axis, black dots), the plant gains Pi. At a relative ATP-release and Pi-production of 1 (gray triangles), the plant release as much phosphate as it takes up, while at higher ATP-release values, the plant loses Pi.

D   Schematic representation of the fluxes for three scenarios: (i) no ATP-release and Pi-production (left, black dots in A–C); (ii) moderate ATP-release, adenosine (A), and Pi-production (middle, gray triangles in A–C); and (iii) high ATP-release, A, and Pi-production (right, white squares in A–C). If there is no ATP-release (left), there is a constant flux of Pi from the fungus via the apoplast to the plant and a constant flux of sugars in the inverse direction. The energy for these fluxes is provided by the phosphate gradient between fungus and plant and the sugar gradient between plant and fungus. At a moderate ATP-release and Pi-production rate (middle, value 1 on the *x*-axis, gray triangles in A–C), there is no Pi-flux from the fungus to the apoplast anymore. The unchanged uptake of sugars is now energized by the proton pump of the fungus. In this condition, the plant retrieves the cleaved Pi originating from the ATP-release via the H/P cotransporter, a transport which is energized by both the sugar gradient and the proton pump. At a high ATP-release and Pi-production rate (right, value 2 on the *x*-axis, white squares in A–C), there is a larger Pi-uptake by the plant and a Pi-uptake by the fungus. Both processes are energized by larger activities of the proton pumps.

and its presence at the host cell surface needs to be proven. Yet, it is tempting to speculate that a similar mechanism is also in place in plants where metabolites derived from the hydrolysis of extracellular nucleotides modulate eATP-triggered immunity once the signaling is no longer required.

Taken together, our work shows that eATP perception plays an important role in plant–*S. indica* interaction in the roots and that *S. indica* has an intrinsic mechanism to modulate bioactive extracellular nucleotide levels in the apoplast (Fig 6). It remains to be proven if the effects mediated by *S. indica* ecto-5′-nucleotidase *in planta* are due to the depletion of the signaling molecule eATP or to accumulation of hydrolysis products, such as adenosine, a potent

immune suppressor in animal systems, and/or phosphate, a regulator of compatibility in mycorrhizal associations. The contribution of a potential immunomodulatory agent derived from the action of the *Si*E5′NT is supported by the reduced response of the marker gene At1g58420 in the *Si*E5′NT plants compared to the *dorn1-3* mutant plants. Additionally, in support of the hypothesis that hydrolysis products may mediate colonization, Daumann *et al* [57] have shown in a recent study that accumulation of adenosine in the apoplast of *Arabidopsis* can promote pathogenic fungal infection in plants. Metabolomics data obtained from APF of colonized roots by sebacinoid fungi showed an accumulation of adenosine at 6 dpi compared to non-colonized roots (Gerd Balcke, Hanna Rövenich,

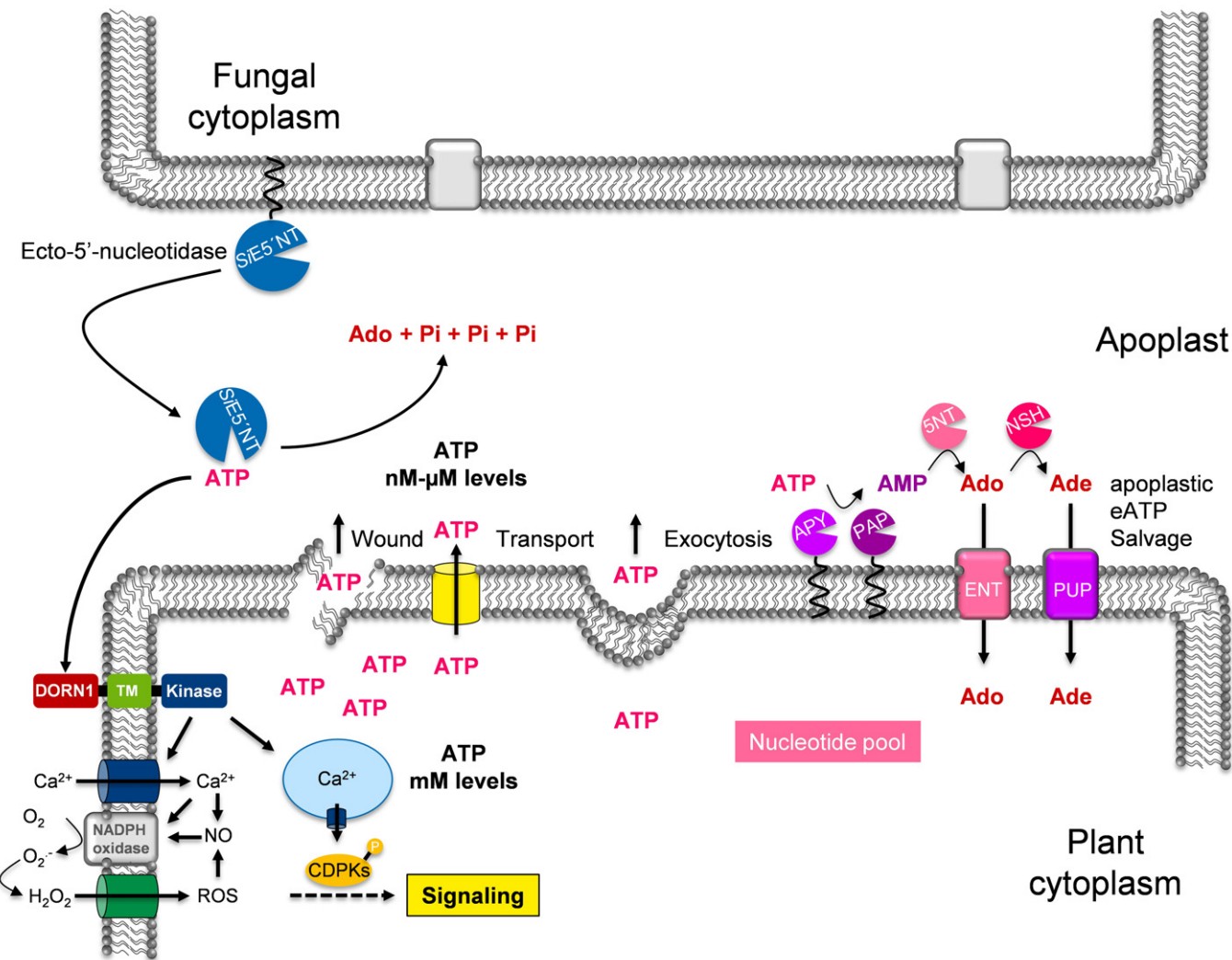

**Figure 6.  Schematic model showing the potential interference of *Si*E5′NT with apoplastic eATP signaling and nucleotide salvage pathway in *Arabidopsis*.**
The schema was modified from Ref. [5]. ENT = equilibrative nucleoside transporters, PAP = purple acid phosphatases, Apy = ectoapyrases or nucleoside triphosphate diphosphohydrolases (NTPDases), NSH = nucleoside hydrolases, 5NT = hypothetical 5′-nucleotidases, PUP = hypothetical purine permeases.

and Alga Zuccaro, personal communication). Thus, it would be important to analyze the role of purine nucleosides in plant root–fungus interactions. Although our data indicate that the activity of the *Si*E5′NT is important for its function in the apoplast, we cannot fully exclude the possibility that an additional function independent of the hydrolytic activity is responsible for the effects observed in this study. Enzymatic inactivation by targeted point mutations in the catalytic domain could help address this possibility.

Modeling of phosphorus/carbon exchange between the plant and the fungus in the presence of an apoplastic Pi-source derived from eATP hydrolysis suggests that in the early phase of interaction, the fungus could reduce phosphate transfer to the plant without affecting the carbohydrate flux from the plant and eventually use this phosphate as a nutrient to support its own growth. Secreted E5′NT homologues are present also in fungal pathogens as *C. incanum* and other endophytes of *Arabidopsis*, such as *C. tofieldiae*, where their expression is induced during colonization

[50]. Fungal genes encoding enzymes involved in ATP-scavenging were also found in the apoplastic fluid of rice leaves infected with the rice pathogens *M. oryzae* and *C. miyabeanus* [43,58], suggesting that extracellular purine-based biomolecules play a role also in other plant–fungus interactions. Hydrolysis of plant-derived primary metabolites with immune modulating functions in the apoplast could thus represent a general mechanism of plant-associated fungi in modulating plant nucleotides signaling as well as serving nutritional needs.

## Materials and Methods

### Fungal strains and culturing techniques

*Serendipita indica* (syn. *Piriformospora indica*, DSM11827, Leibniz Institute DSMZ—German Collection of Microorganisms and Cell

Cultures, Braunschweig, Germany) and *Serendipita indica* GoGFP strain expressing cytosolic GFP [36] were grown in liquid CM supplemented with 2% glucose and incubated at 28°C with constant agitation at 120 rpm, or on solid medium supplemented with 1.5% agar [59]. *Ustilago maydis* SG200 [60] cultures were propagated on potato dextrose agar (Difco) or YEPS<sub>Light</sub> liquid medium at 28°C [61]. *Colletotrichum incanum* strain MAFF238704 was kindly provided by Prof. Paul Schulze-Lefert, MPIPZ, Cologne, and propagated on solid CM supplemented with 1.5% agar in darkness at 25°C [50].

**Plant growth conditions and fungal inoculation**

Barley seeds (*Hordeum vulgare* cv. Golden Promise) were surface sterilized as described previously [36] and germinated for 3 days in the dark on wet filter paper. Germlings were transferred into jars with 1/10 PNM agar [59] and inoculated with 3 ml (500,000 spores/ml) *S. indica* chlamydospores in 0.002% Tween-20 aqueous solution and cultivated in a growth chamber with a day : night cycle of 16 h : 8 h (light intensity 108 µmol/m$^2$/s and temperature of 22°C:18°C). Tween water-treated germlings were used as control. Three independent biological replicates were used. For each biological replicate, four to five barley seedlings in one jar were pooled at each time point.

*Arabidopsis* seeds were surface sterilized with 70% ethanol for 10 min followed by 100% ethanol for 7 min and then air-dried. The sterilized seeds were transferred to half-strength Murashige and Skoog (½ MS) medium containing 1% sucrose and 0.4% Gelrite (Duchefa), incubated for 3 days in the dark at 4°C, and subsequently grown for 7 days with a day : night cycle of 8 h : 16 h (light intensity, 47 µmol/m$^2$/s) at 24°C. Three independent biological replicates were used for all *Arabidopsis* experiments unless otherwise stated in the legend. For each biological replicate, three plates containing 20 plants each were pooled. For colonization studies, 500,000 spores/ml of *S. indica* chlamydospores in 0.002% Tween-20 aqueous solution was applied to the roots of each plant of 7-day-old seedlings. Tween water-treated seedlings were used as control.

For colonization analysis using *C. incanum, Arabidopsis* seedlings were grown for 10 days under a day: night cycle of 8-h: 16-h regime (light intensity 111 µmol/m$^2$/s and temperature of 22°C:18°C). Roots were drop-inoculated with 200,000 spores/ml of *C. incanum* suspended in sterile distilled water. Water-treated seedlings served as control. Three independent biological replicates were used each representing a pool of roots from 20 plants.

**Identification of *S. indica* proteins in apoplastic fluid and culture filtrate**

Barley seeds (*Hordeum vulgare* L. cv Golden Promise) were surface sterilized for 1 min under light shaking in 70% ethanol followed by 1 h in 12% sodium hypochlorite. Subsequently, seeds were washed repeatedly for 1 h with sterile distilled water. After germination for 3 days at 22°C in complete darkness on wet filter paper, germinated seeds were transferred into jars with 1/10 PNM [59] under sterile conditions and inoculated with 3 ml of chlamydospores (concentration 500,000 spores/ml) of *S. indica* GoGFP strain [36] and cultivated in a growth chamber with a day : night cycle of 16 h: 8 h (light intensity, 108 µmol/m$^2$/s) and temperature of 22°C:18°C. For

APF extraction, three time points reflecting different interaction stages have been chosen: biotrophic phase (5 dpi) as well as cell death-associated phase (10 and 14 dpi). After respective growth time, roots from 300 seedlings for each treatment and time point were thoroughly washed and cut into pieces of 2 cm length (2$^{nd}$ to 4$^{th}$ cm of the root). Samples were vacuum-infiltrated (Vacuubrand, CVC 3000, VWR) with deionized water 5 × 15 min at 250 mbar with 1-min atmospheric pressure break. Bundled infiltrated roots were centrifuged in 5-ml syringe barrels at 2,000 rpm in a swing bucket rotor (800 *g*) for 15 min at 4°C as described in Ref. [25]. Pooled APF samples were stored at −20°C. In order to exclude cytoplasmic contamination, apoplastic fluid samples were tested in immunoblots with GFP antibody. For Western blotting, proteins from unstained gels were transferred onto nitrocellulose membranes using a semidry blotting system from Bio-Rad (Hercules, USA). Anti-GFP antibody was obtained from Cell Signaling Technology (Danver, USA) and used at a 1:2,000 dilution. An anti-mouse antibody purchased from Sigma (Munich, Germany) was used as secondary antibody at a dilution of 1:2,000. For higher peptide mass accuracy, part of the APF samples was deglycosylated with Protein Deglycosylation Mix (P6039S, New England Biolabs, Ipswich, USA) under denaturing conditions. In agreement with Ref. [62], at the centrifugation force used the samples were free from obvious symplastic contamination and were thus regarded as being of apoplastic origin.

Secreted proteins from culture filtrate were prepared as follows: *S. indica* cultures grown for 7 days in liquid CM were passed through a miracloth filter (Merck Millipore, Darmstadt), a folding filter type 600P (Carl Roth, Karlsruhe, Germany), and finally a 0.45-µm syringe filter (Millex-GP; Merck Millipore, Darmstadt, Germany) to remove small mycelium fragments and spores. Proteins were precipitated by addition of 10% (v/v, final concentration) trichloroacetic acid and incubation at −20°C overnight and subsequent centrifugation at 40,000 *g*. For SDS–PAGE, pelleted secreted proteins resolved in 50 µl of 1× SDS sample buffer and 15 µl were used. For mass spectrometric analysis, pelleted secreted proteins were solved in 50 µl TBS (50 mM Tris pH 7.5, 150 mM NaCl). Half of the samples were deglycosylated with the Protein Deglycosylation Mix (New England Biolabs, Ipswich, USA) under denaturing conditions according to manufacturer's protocol. Proteins were then analyzed via liquid chromatography–electron spray ionization–tandem mass spectrometry (LC-ESI-MS/MS; LTQ Orbitrap Discovery; Thermo Fisher Scientific, Waltham, USA) after tryptic digest (FASP™ Protein Digestion Kit; Expedeon, Swavesey, UK) in the facilities of CEACAD/CMMC Proteomics Facility in Cologne, Germany. Peptide masses were compared to *S. indica in silico* trypsin-digested proteome (Genbank, NCBI) with an inclusion of GFP, with a barley dataset downloaded from the IPK Gatersleben homepage (http://webblast.ipk-gatersleben.de/barley_ibsc/downloads/) and with *Rhizobium radiobacter* F4 (syn. *Agrobacterium tumefaciens*, syn. *Agrobacterium fabrum*) dataset downloaded from Genbank, NCBI (WGS project JZLL01000000, https://www.uniprot.org/proteomes/UP000033487). The exact instrument settings can be found in the Supplementary Appendix containing the processed data. Mass spectrometric raw data were processed with Proteome Discoverer (v1.4, Thermo Scientific) and SEQUEST. Briefly, MS2 spectra were searched against the respective databases, including a list of common contaminants. The minimal peptide length was set to seven amino acids, and carbamidomethylation at cysteine residues was considered as a fixed modification. Oxidation (M) and Acetyl (Protein

N-term) were included as variable modifications. The match-between-runs option was enabled. Fungal-derived proteins matched to peptides were analyzed for the presence of signal peptides and exclusiveness for the three specific time points of APF extraction.

## Construction of plasmids

Full-length *E5′NT* (PIIN_01005) was amplified from cDNA of barley roots inoculated with *S. indica* at 5 dpi by respective oligonucleotides (Table EV5) and cloned into the TOPO® cloning plasmid (Thermo Fisher Scientific, Waltham, USA) resulting in TOPO-E5′NT. For the expression of various *E5′NT* constructs under the control of the synthetic otef promoter which exhibits strong constitutive expression [63], full-length *E5′NT, E5′NTwoGPI, and E5′NTwoSPwoGPI* were amplified from the plasmid TOPO-E5′NT using Phusion DNA polymerase (NEB) by respective oligonucleotides (Table EV5) and cloned into the *Bam*HI-*Not*I digested plasmid $^{Potef::}$Yup1(RFP)$_2$ [64] using Gibson assembly [65], generating the plasmids $^{Potef::}$E5′NT, $^{Potef::}$E5′NT$^{woGPI}$, and $^{Potef::}$E5′NT$^{woSPwoGPI}$, respectively.

For the construction of SP$_{E5′NT}$:mCherry:E5′NT$^{woSP}$, *Si*E5′NT without signal peptide (*E5′NT$^{woSP}$*) was amplified from the plasmid TOPO-E5′NT using Phusion DNA polymerase (NEB) by respective oligonucleotides (Table EV5) and cloned into the *Bam*HI digested plasmid $^{ProUmPit2::}$SP$_{Dld1}$:mCherry:Dld1$^{woSP}$ (Nostadt *et al*, unpublished data), generating the plasmid $^{ProUmPit2::}$SP$_{Dld1}$:mCherry:E5′NT$^{woSP}$. Subsequently, $^{ProUmPit2::}$SP$_{Dld1}$:mCherry:E5′NT$^{woSP}$ was digested with *Sac*II and *Nco*I and ligated with a DNA fragment encoding the E5′NT signal peptide (SP$_{E5′NT}$) digested with the same restriction enzymes, replacing the Dld1 signal peptide (SP$_{Dld1}$) and generating the construct $^{ProUmPit2::}$SP$_{E5′NT}$:mCherry:E5′NT$^{woSP}$. For expression in *A. thaliana* under the control of CaMV 35S promoter (Pro35S), full-length *E5′NT*, SP$_{E5′NT}$:mCherry:E5′NT$^{woSP}$, and mCherry were amplified from the plasmids TOPO-E5′NT and $^{ProUmPit2::SP}$$_{E5′NT}$:mCherry:E5′NT$^{woSP}$, respectively, using Phusion DNA polymerase (NEB) by respective oligonucleotides (Table EV5) and cloned into the *Bam*HI digested plasmid pCXSN [66], using Gibson assembly, generating the plasmids $^{Pro35S::}$E5′NT, $^{Pro35S::}$SP$_{E5′NT}$:mCherry:E5′NT$^{woSP}$, and $^{Pro35S::}$mCherry, respectively.

## *Ustilago maydis* transformation

The *U. maydis* strains were transformed by integration of the p123-derived plasmids into the ip locus in the haploid solopathogenic strain SG200 as previously described [67].

## *Ustilago maydis* cell surface 5′-nucleotidase activity assay

For the cell surface 5′-nucleotidase activity assay, $^{Potef::}$E5′NT, $^{Potef::}$E5′NT$^{woGPI}$ $^{Potef::}$E5′NT$^{woSPwoGPI}$ along with the control $^{Potef::}$mCherry *U. maydis* strains were grown overnight in YEPS$_{Light}$. Then, the cultures were diluted in PD broth and grown until OD$_{600}$ reached 0.8. The cells were harvested through centrifugation and washed two times with 0.9% NaCl and finally re-suspended in equal volume of sterile double deionized water. Then 100 μl of cell suspension from each *U. maydis* strains was incubated with 50 μM of ATP, ADP, or AMP for 30 min, and the hydrolysis of these nucleotides was measured by quantifying the amount of phosphate

released using EnzChek® Phosphate Assay Kit (Thermo Fisher Scientific).

## *Arabidopsis thaliana* transformation

The plasmids $^{Pro35S::}$E5′NT (#303), $^{Pro35S::}$SP$_{E5′NT}$:mCherry:E5′NT$^{woSP}$ (#304), and $^{Pro35S::}$mCherry (#305) were transformed into the *Agrobacterium* strain GV3101 by electroporation using Gene Pulser Xcell Electroporation System (Bio-Rad Laboratories, Hercules, CA, USA) following the manufacturer indications. The cells were plated on LB medium containing 25 μg/ml of rifampicin and 50 μg/ml of kanamycin. *Agrobacterium*-mediated transformation of *A. thaliana* was carried out by means of the floral dip method [68,69]. Stratified seeds from T0 plants were grown in ½ MS containing 30 mg/l hygromycin under short day condition. Putative transgenic lines were selected by their hygromycin resistance. Putative transgenic lines were transferred to soil and 2 weeks later tested by PCR using a gene-specific primer (Table EV5). The T$_3$ transgenic lines from $^{35Spro::}$E5′NT (#303) and $^{35Spro::}$SP$_{E5′NT}$:mCherry:E5′NT$^{woSP}$ (#304) were confirmed again by both hygromycin resistance and PCR using a gene-specific primer. In addition, the elevated expression level of *E5′NT* in $^{35Spro::}$E5′NT (#303) and in $^{35Spro::}$SP$_{E5′NT}$:mCherry:E5′NT$^{woSP}$ (#304) as well as the *mCherry* in $^{35Spro::}$mCherry (#305) lines was analyzed by quantitative real-time PCR. The transgenic lines which had the highest transcript levels were selected for further studies.

## ATP assay

To test whether colonization of *S. indica* in *Arabidopsis* and barley triggers the release of eATP, 7-day-old *Arabidopsis* seedlings and 3-day-old barley seedlings were treated either with a *S. indica* spore suspension in 0.01% tween water (5 × 10$^5$ spores/ml) or with 0.01% tween water for 2.5 h. The *Arabidopsis* treated seedlings were transferred to 60-mm petri plates (3 plates per treatment). Each plate contained 3 ml of liquid ½ MS (ammonium-free) medium and 15 treated seedlings. Subsequently, seedlings were grown for 3, 5, 7, and 10 days postinoculation (dpi) with constant light at 22°C. Three individual samples of 50-μl media were collected from each plate, flash frozen in liquid nitrogen, and stored in −80°C until eATP measurements were carried out. Apoplastic fluid from colonized and non-colonized barley roots was collected as mentioned above, flash frozen in liquid nitrogen, and stored in −80°C until eATP measurements were carried out. The measurement of eATP in the growth media and in the APF was performed using the ENLITEN ATP Assay System from Promega. Reactions were carried out in 96-well plates, and luminescence was detected using a microplate reader (TECAN Safire).

## Membrane protein purification

Membrane-bound proteins were extracted from 1 g shoot material from the respective *A. thaliana* lines (25 days old) grown in soil. The material was ground in liquid nitrogen and re-suspended in 5 ml 0.05 M Tris–HCl buffer containing 1 mM MgCl$_2$ (pH 7.5). After centrifugation for 20 min at 12,000 *g*, the pellet was re-suspended in 5 ml Tris buffer containing 1% IGEPAL CA-630 (Sigma), gently stirred at 4°C for 30 min, and subsequently centrifuged at 5,000 *g*

for 10 min. The supernatant was cleared by ultracentrifugation at 100,000 *g* for 45 min. The resulting supernatant was used for subsequent enzyme activity assays.

## Measurement of E5′NT and apyrase activity assay in *A. thaliana* lines

The activity of purified, membrane-bound apyrases and the overexpressed *Si*E5′NT was monitored using the EnzChek® Phosphate Assay Kit (Thermo Fisher Scientific) according to manufacturer's instruction. 15 μl of purified membrane proteins was used for the enzyme kinetics assay. 100 μM ATP, ADP, or AMP (in 10 mM Tris/MES buffer containing 2 mM $MgCl_2$ and 30 mM KCl; pH 6.5) were used as substrates. The background absorbance at 360 nm was monitored for 5 min before the addition of the substrates. Subsequently, kinetics were measured for 2 h.

## Real-time PCR analysis

DNA isolation was performed from ~200 mg of ground material. The powder was incubated for 5 min at RT under slight rotation with ~500 μl CTAB buffer (2% hexadecyl trimethyl-ammonium bromide, 100 mM Tris–HCl, 20 mM EDTA, 1.4 M NaCl, and 1% polyvinyl pyrrolidone vinylpyrrolidine homopolymer Mw 40,000, pH 5.0). Subsequently, 250 μl chloroform : isoamyl-alcohol (24:1) was added followed by an additional incubation for 5 min at RT and centrifugation. The water phase was collected, and polysaccharides were precipitated using ethanol before final DNA precipitation using 1 volume of isopropanol. Total RNA was isolated from 200 mg of ground fungal material or colonized and non-colonized plant root material using the TRIzol® reagent (Invitrogen, USA) as described previously [36]. Contaminating genomic DNA was removed by DNase I treatments (Thermo Fisher Scientific, USA) according to manufacturer's protocol. First-strand cDNA was synthesized with 1 μg of total RNA primed with Oligo-dT and random hexamer using First Strand cDNA Synthesis Kit (Thermo Fisher Scientific, USA). The resulting cDNA preparations were diluted to a final concentration of 2.5 ng/μl, and 4 μl of each cDNA sample was used for qRT–PCR. *Serendipita indica* translation elongation factor 1 (*SiTEF*) and *A. thaliana SAND* or *UBI* gene primers were used for qPCR amplifications. The reactions were performed in 96-well reaction plate (Bio-Rad, USA) on a Real-Time PCR System (Bio-Rad, USA). Each well contained 7.5 μl of 2× SYBR® Green QPCR Master Mix (Promega, USA), 4 μl of cDNA or 10 ng of DNA, and 0.7 μl of each primer from a 10 μM stock, in a final volume of 15 μl. Transcripts of each gene were amplified using the primers described in Table EV5. The threshold cycles ($C_T$) of each gene were averaged from three replicate reactions, and the relative expression or plant to fungus ratio from gDNA was calculated using the $2^{-\Delta CT}$ method [70].

## E5′NT expression levels in axenic culture in response to nucleotides

In order to test whether *Si*E5′NT expression is induced in the presence of nucleotides, WT *S. indica* spores (500,000 spores/ml) were used to inoculate liquid CM medium and grown for 5 days at 28°C with 110 rpm shaking. Subsequently, the culture was filtrated through miracloth, and the mycelium was crushed and regenerated

in 100 ml fresh CM medium for 2 days. After regeneration, 5 ml of fungal suspension was diluted with 45 ml of CM medium supplemented with 100 μM of the respective nucleotides. At the indicated time points, the mycelium was harvested by filtration, washed with water, and flash frozen in liquid nitrogen before processing according to the Real-Time PCR Analysis Protocol listed above. Three to six independent biological replicates for each time point/nucleotide combination were performed.

## Calcium influx quantification

All calcium influx experiments were conducted using plants that ectopically express aequorin as $Ca^{2+}$ reporter (Choi *et al* 2014). Seven-day-old seedlings grown on ½ MS plates containing 1% sucrose were transferred to 96-well plates containing 50 μl 10 μM coelenterazine (Promega) in reconstitution buffer (2 mM MES/KOH containing 10 mM $CaCl_2$; pH 5.7). After 16 h, the luminescence background was measured for 5 min followed by the addition of different adenylates (final concentration 100 μM ATP, ADP, AMP, or adenosine). Luminescence was monitored for 30 min.

## Confocal microscopy

Confocal images were recorded using a TCS-SP8 confocal microscope (Leica, Bensheim, Germany). mCherry fluorescence was imaged with an excitation of 561 nm and a detection bandwidth set to 580–630 nm. DAPI and auto-fluorescence of the samples were assessed by laser light excitation at 405 nm with a detection bandwidth set to 415–465 nm.

## Homology modeling of *Si*E5′NT

In order to carry out the homology modeling of *Si*E5′NT, best template was selected through PSI BLAST against the PDB database (http://www.rcsb.org/pdb/home/home.do). The best hit with high sequence identity and atomic resolution < 1.8 Angstroms was selected as template. The three-dimensional structure of *Si*E5′NT was generated using a restrained-based approach in MODELLER 9v11 [71]. Initial refinement of the 3D model generated was carried out with the help of loop refinement protocol of MODELLER. The assessment of the final structural model was carried out with PROCHECK [72], ProSA [73], and QMEAN [74] analyses. The structure was visualized using PyMOL (http://www.pymol.org/).

## Computational cell biology

The nutrient exchange of phosphates (Pi) and sugars (C) between plant and fungus was essentially performed as described previously in mycorrhizal interactions [53], in a minimal model combining proton-coupled phosphate (H/P) and sugar (H/C) transporters and proton pump-driven background conductances. The minimal model was enlarged by adding the release of ATP from the plant into the apoplast and its subsequent cleavage into phosphates and adenosine (A) by E5′NT enzymes. The Pi and C exchange between the plant and the fungus was simulated for different values of ATP-release and Pi-production. The ATP-release- and Pi-production rate were normalized that at the relative value of 1, the fungus does not export Pi. The mathematical description of these processes results in

redundant parameters, which were combined. Therefore, ATP-release and cleavage were considered as one process in this study, which was modeled as a constant Pi-production rate: $d[Pi]_{apo} = const \times dt$. Following the mathematical description of all transporters, an *in silico* cellular system was programmed and computational cell biology (dry-lab) experiments were performed using the VCell Modeling and Analysis platform developed by the National Resource for Cell Analysis and Modeling, University of Connecticut Health Center [75].

**Expanded View** for this article is available online.

## Acknowledgements

We are indebted to Prof. Gary Stacey for providing the *A. thaliana dorn1-3* mutant and Col-0$^{aq}$ lines. Heidi Widmer is acknowledged for her excellent support in the laboratory. Ganga Jeena is acknowledged for her support with bioinformatics analysis. We are also grateful to Dr. Hanna Rövenich for her critical reading of the manuscript. ID was supported by the Fondecyt grant No. 1150054 of the Comisión Nacional de Investigación Científica y Tecnológica (CONICYT) of Chile. AZ acknowledge support from the Cluster of Excellence on Plant Science (CEPLAS, EXC 1028), DFG ZU 263/2-1, DFG ZU 263/3-1, and institutional funds of the Max Planck Society.

## Author contributions

AZ and SN conceived the project, designed the experiments, and wrote the paper with contributions from SW and GL. SN, SW, XQ, RN, FG, FS, and GL performed experiments and analyzed the data. ID produced the P/C model.

## Conflict of interest

The authors declare that they have no conflict of interest.

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
