## [Review Process File · EMBO Reports]

***Serendipita indica* E5'NT modulates extracellular nucleotide levels in the plant apoplast and affects fungal colonization**

Shadab Nizam, Xiaoyu Qiang, Stephan Wawra, Robin Nostadt, Felix Getzke, Florian Schwanke, Ingo Dreyer, Gregor Langen, Alga Zuccaro

Review timeline:	Submission date:	20 November 2018
	Editorial Decision:	29 November 2018
	Revision received:	5 December 2018
	Editorial Decision:	11 December 2018
	Revision received:	13 December 2018
	Accepted:	14 December 2018

Editor: Esther Schnapp

Transaction Report: This manuscript was transferred to *EMBO reports* following peer review at *The EMBO Journal*

1st Editorial Decision

29 November 2018

Thank you for the transfer of your revised manuscript to EMBO reports. I have now looked at all files, and a few changes are still necessary.

As far as I understand, the manuscript does not provide direct data that show an effect on the plant immune response. This was also mentioned by the referees. The response shown is to eATP. I therefore think that the statements on the plant immune response in the title, abstract and discussion (when referring to Fig 5) need to be toned down.

The manuscript has 5 main figures but separate results and discussion sections. Either one more main figure needs to be added/moved from the supplementary figures, or the results and discussion sections need to be combined. This is the difference between our scientific reports and full articles. You can find more information in our guide to authors online.

Fig 2A does not specify the p-value and tests, Fig 4E does not specify "n" and the error bars, SF6 does not specify the bars and error bars, and SF8C mentions n=2 in which case no error bars can be shown. Fig 8E needs to explain the box plots.

Tables S1-S5 should either be EV tables or Datasets (named Table EV1, 2, or Dataset EV1, 2, etc). All tables need titles and can have legends. All tables need to be uploaded as individual files, either word or excel. The Appendix table 1 also needs a title and may be legend, and it can also be either an EV table or a Dataset. A Dataset and EV table can have more tabs, but a regular table cannot.

The Appendix file needs to be a single file with table of content and the figure legends included. The nomenclature is Appendix Figure S1, Appendix Table S1, etc. Only the EV figure legends are part of the main manuscript file. Fig S11 in the Appendix is probably also a Dataset. The methods in the Appendix file should only stay there if they exclusively refer to data shown in the Appendix file. Otherwise they should be moved to the main methods.

The text in the figures is often too small to be read. All text needs to be readable. You can check our

figure guidelines here:

http://www.embopress.org/sites/default/files/EMBOPress_Figure_Guidelines_061115.pdf

You mention "data not shown" on page 10 and 13, which we don't allow. Please either remove the statement or may be use "personal communication".

Section B in the Checklist on statistics needs to be completed.

EMBO press papers are accompanied online by A) a short (1-2 sentences) summary of the findings and their significance, B) 2-3 bullet points highlighting key results and C) a synopsis image that is 550x200-400 pixels large (the height is variable). You can either show a model or key data in the synopsis image. Please note that text needs to be readable at the final size. Please send us this information along with the revised manuscript.

I look forward to seeing a final manuscript as soon as possible. Please let me know if you have any questions or comments.

REFEREE REPORTS

Referee #1:

This manuscript investigates the role of eATP as a signal molecule interactions between the mutualistic fungus *S. indica* and its host plants. First an LC-MS/MS analysis of apoplastic fluid from infected barley plants was used to catalogue proteins and identify a fungal ecto-nucleotidase as a candidate effector influencing the level of eATP to modulate the plants immune response. Secondly the eATP levels in the apoplast were measured during *S. indica* colonization of Arabidopsis and barley and the level of ecto-nucleotidase estimated in axenic cultures and in plants. To investigate the function of the ecto-nucleotidase in more detail the full length *S. indica* ecto-nucleotidase gene was then expressed in Arabidopsis and the *Ustilago maydis* fungal pathogen in order to measure the ATP, ADP, AMP hydrolyzing capacity as well as the *S. indica* infection rate. Higher levels of ATP hydrolysis associated with Arabidopsis membrane fractions correlated with increased colonization and reduced eATP levels. Finally, expression of previous identified eATP responsive genes were measured by qPCR.

The involvement of effectors during colonization by mutualistic fungi is relatively unknown. This study identifies an extracellular ecto-nucleotidase and investigates the function of this effector in modulation of eATP in the apoplast and effects on immunity. The function of eATP has mainly been studied in pathogen infection. In this perspective the novelty of the work presented in the manuscript is in extending the eATP biology to mutualistic fungi and in identifying a fungal effector ecto-nucleotidase.

Major concerns:

The eATP in Arabidopsis is estimated from the level in the surrounding medium while in barley the apoplastic fluid is isolated and the eATP measured. One would think that the level in the apoplastic fluid is a more precise methods and there is no explanation for the use of two different methods.

This inconsistency is carried into the measurements of ecto-nucleotidase activity and eATP in the transgenic Ecto-5^{NT} lines. Here another method, membrane associated hydrolytic activity, is used while eATP is again measured in the medium. This is confusing and it would be more convincing and consistent to show results from apoplastic fluids, which is where the process is supposed and suggested to be localized.

One of the conclusions from this work is that immune signaling is influenced by the ecto-nucleotidase activity and expression of already known eATP response genes is measured to support this conclusion. However, immune responses in Arabidopsis are very well characterized and it is surprising that expression of central genes or even mutant normally used in immunity studies are not

included in the manuscript.

The discussion of the results is not informative and results and conclusion obtained are not discussed in relation to current knowledge in plant-microbe interaction and immunity. Lines 335 - 393 is a review type summary and expression of the authors beliefs. Line 394 -423 just repeats observation reported in the main text.

Minor concerns:

The modelling of Pi-source effects, line 311- 333 is not useful without experimental support, as stands is appears like a speculative add-on.

Nomenclature used for constructs for example "Potef::mCherry, Potef::E5'NTwoGPI and Potef::E5'NTwoSPwoGPI" is clumsy and almost unreadable.

Referee #2:

In the present study, the authors report LC-MS/MS proteomics studies on the apoplasmic fluid of barley roots following colonization with the beneficial fungus *S. indica*. Of the identified 102 fungal proteins, the authors put a particular focus on an ecto-5'-nucleotidase (E5NT) among several ATP-scavenging enzymes that are specifically present in the fungal colonized roots. The authors further show that eATP levels are increased in response to *S. indica* colonization in barley and Arabidopsis roots, and that the eATP receptor DORN1 restricts fungal colonization. Based on sequence alignment and structure prediction, the authors suggest that the fungal E5NT acts as a monomer and hydrolyzes different nucleotides. In ex vivo expression systems in Arabidopsis and in *Ustilago maydis*, the introduction of SiE5NT exhibits a significant increase in the hydrolysis activity for ATP, ADP and AMP. Moreover, the authors show that transgenic expression of SiE5NT in Arabidopsis results in enhanced colonization with *S. indica* and another pathogen, *C. incanum*, in line with a decrease in eATP levels. Based on these results, the authors propose a model in which plant-inhabiting fungi, including *S. indica*, secrete eATP-hydrolyzing enzymes to dampen an important branch of DAMP-mediated host immunity for promoting fungal colonization. Overall, this work has the potential to provide significant insight into beneficial plant-microbe associations and colonization strategies of plant growth-promoting and pathogenic fungi, important topics that attract high attention from the society. However, in my view, this work is incomplete in the following aspects.

- 1) There is not direct evidence that E5NT-mediated decrease of eATP levels suppresses DORN-mediated defense responses. It remains not clear whether the observed degree of decrease in eATP levels is sufficient to attenuate DORN1 signaling or whether the eATP pools monitored by DORN1 are influenced by E5NT in the transgenic plants or during fungal colonization. It would help if the authors test E5NT effects on defense responses to exogenous eATP application.
- 2) As the authors mention in the text, changes in eATP levels can influence different pathways in plants. It is important to determine whether, and if so to which extent, the E5NT-mediated increase of fungal colonization is DORN1 dependent, by testing E5NT introduction (overexpression) effects in the *dorn1* mutant background. This will also help address the above concern (1).
- 3) What is the difference in *S. indica* colonization between Figs 4C and E? The results should be the same but very much different for the same lines used.
- 4) It also remains to be addressed whether and how increased *S. indica* colonization when eATP-triggered defenses are compromised influences different benefits of the fungal association. Are the previously described *S. indica* benefits reduced or increased in E5NT-expressing transgenic plants and *dorn1* mutant plants? Induction of defense marker genes following *S. indica* inoculation was rather increased when eATP-triggered defenses were compromised, which makes me wonder what happens to the plant-fungus association afterward.
- 5) In principle, the substrate specificity of the E5NT protein needs to be tested in vitro by using recombinant protein purified from the cells. Otherwise, it is better to tone down discussions regarding the substrate specificity.
- 6) To raise the hypothesis that eATP hydrolysis also contributes to nutrition, the authors should

discuss possible co-expression of e.g. phosphate transporter genes with E5NT.

Detailed comments.

Lines 84-87: The logical flow seems to be abrupt. Was there any link previously known between *S. indica* and eATP-mediated immunity? If so, please explain it.

Table S1 is missing (which could be my fault).

Figure legends should be more kind and self-explanatory. For instance, what are S1, S2d or S3d etc. in Fig 1B? When were the fungal inoculation data obtained (how many dpi)?

Referee #3:

In this comprehensive study Nizam et al. describe 1) a protein MS-based survey of the barley root apoplast during colonization with the beneficial and root-associated fungus *Serendipita indica*, indicating an enrichment of apoplastic fungal proteins related to purine metabolism 2) report a drastic increase in eATP levels at early stages of fungal host colonization in the barley and *A. thaliana* root apoplast as well as a hypercolonization phenotype of *A. thaliana* *dorn1* mutant roots known to be impaired in the perception of eATP. The authors then have 3) chosen one of the secreted potentially ATP-scavenging enzymes of *S. indica*, denoted PIIN_01005, for in-depth functional analysis. They demonstrate 4) evidence for ecto-5'-nucleotidase activity by heterologous expression of *S. indica* PIIN_01005 in the Basidiomycete *Ustilago maydis* and *A. thaliana* transgenic plants using washed culture suspensions and enriched plasma membrane fractions, respectively. 5) A *S. indica* hypercolonization phenotype is reported in *A. thaliana* transgenic lines constitutively expressing PIIN_01005 and this 6) correlates with a slight reduction (~10%) in eATP levels at 5 days post inoculation (dpi). 7) An enhanced disease susceptibility phenotype is found in the *A. thaliana* transgenic line constitutively expressing PIIN_01005 upon inoculation with the root fungal pathogen *Colletotrichum incanum*. Finally, 8) metabolic modelling was applied to infer that eATP hydrolysis might serve a role in Pi nutrition for the fungus and in modulating plant innate immunity.

Very little is known about the functions of eATP during colonization with pathogenic and beneficial microorganisms. Thus, the present study has the potential to significantly advance our knowledge in this understudied research area in plant-microbe interactions. Overall the manuscript is well written and the Figures are organized in a logical sequence. The first two results sections describing the protein MS-based analysis of the root apoplast as well as *S. indica*-induced massive increases in eATP levels are largely solid and impressive (Fig. 1 and Fig. 2).

My major concern(s) with this work begins with the functional analysis of *S. indica* PIIN_01005, all inferences made from Fig. 5 plus the corresponding supplementary Figures as well as with conclusions based on the metabolic modelling (Fig. S10) as listed below. Unfortunately, the results corresponding to Fig. 5 are central for the major claim of this study and additional experimentation is critical to convince this reviewer. In addition, the Method section is in several cases fragmentary so that it is difficult to reconstruct how some of the experiments were conducted, how data were acquired and/or were processed. Below I have described my major points of critique.

1) GO-term enrichment analysis indicated an 'overrepresentation of apoplastic *S. indica* proteins implicated in purine/ATP metabolism' (Table S3, Fig. S1B and S2), suggesting the involvement of diverse extracellular proteins modulating eATP levels. I have difficulties to infer this conclusion from Fig. S2 given the lack of any detailed information in the corresponding Figure caption, Method section, and absence of applied statistical tests with corresponding p-values. Likewise, the authors state that *S. indica* PIIN_01005 is one of the 'most abundant' secreted fungal proteins (ln. 90 to 92). If this conclusion is based on protein MS data, is this supported based on both iBAQ values within samples and LFQ intensities across samples? Ideally, there should be correspondence between these two metrics, but the Method section remains entirely unclear on this point.

2) Key for the major claim of the present study are transgenic *A. thaliana* lines expressing *S. indica* PIIN_01005 encoding SiE5'NT. If my reading of the results section and Figures is correct, then a single *A. thaliana* over-expression line ('L3') was used for quantifying Ecto-5'NT activity and for inoculation experiments with beneficial *S. indica* and pathogenic *C. incanum*. If so, this is

inadequate and evidence from further independent transgenic lines needs to be provided. More importantly, the conclusions of the authors are in my view not justified because critical control transgenic plants expressing a catalytically inactive form of SiE5'NT are lacking. This is necessary to exclude the possibility that the modest reduction seen in eATP levels in the tested transgenic lines at 5 dpi of *S. indica* (Fig. 4D) as well as enhanced *S. indica* root colonization (Fig. 4C) is really the result of ectopic SiE5'NT activity and not due to an unspecific perturbation of root-associated processes independent of SiE5'NT enzyme activity. The data shown with transgenic plants overexpressing mCHERRY are an insufficient control in this context (Fig. 4). Meaningful conclusions are further complicated by the fact that SiE5'NT enzyme activity appears to be compromised in control plants expressing a SiE5'NT-mCHERRY fusion protein (Fig. 4). A flaw in experimental design is the absence of epitope-tagged functional and catalytically inactive SiE5'NT variants expressed in transgenic *A. thaliana*, which would enable the authors to quantify steady-state levels of the plant-expressed SiE5'NT protein in the root apoplast and plasma membrane fractions and relate this to SiE5'NT enzyme activity as well as fungal infection phenotypes. Thus, given the massive increases in eATP levels in the root apoplast at early stages of *S. indica* root colonization of wild-type plants (Fig. 2A and B), I am not convinced that the modest decrease in eATP levels seen upon SiE5'NT overexpression in transgenic plants (Fig. 4D) is causally and directly linked to the observed changes in *S. indica* and *C. incanum* infection phenotypes.

3) Conclusive evidence for SiE5'NT physiological functions also demands the generation of corresponding *S. indica* knock-out mutants and subsequent inoculation experiments with the mutant strains. I understand this is at present technically challenging with *S. indica* and might not be realistic, but the lack of this genetic tool further limits an unambiguous assessment of SiE5'NT function(s) during plant-fungus interactions.

4) Although I am appreciative of metabolic modelling, I have not only difficulties to reconstruct from the method section the calculation of the sparse dataset visualized in Fig. S10, I also believe the conclusion drawn remains speculative unless underpinned by advanced stable isotope labelling experiments. I suggest removing this part from the manuscript as these are premature observations that are not central for the main claim of the work.

Minor point:

Ln. 195: replace 'hypnotized' by 'hypothesized'

Rebuttal

“The fungal root endophyte *Serendipita indica* modifies extracellular nucleotides to modulate plant immunity” by Nizam *et al.*

Referee #1:

This manuscript investigates the role of eATP as a signal molecule interactions between the mutualistic fungus *S. indica* and its host plants. First an LC-MS/MS analysis of apoplastic fluid from infected barley plants was used to catalogue proteins and identify a fungal ecto-nucleotidase as a candidate effector influencing the level of eATP to modulate the plants immune response. Secondly the eATP levels in the apoplast were measured during *S. indica* colonization of *Arabidopsis* and barley and the level of ecto-nucleotidase estimated in axenic cultures and in plants. To investigate the function of the ecto-nucleotidase in more detail the full length *S. indica* ecto-nucleotidase gene was then expressed in *Arabidopsis* and the *Ustilago maydis* fungal pathogen in order to measure the ATP, ADP, AMP hydrolyzing capacity as well as the *S. indica* infection rate. Higher levels of ATP hydrolysis associated with *Arabidopsis* membrane fractions correlated with increased colonization and reduced eATP levels. Finally, expression of previous identified eATP responsive genes were measured by qPCR.

The involvement of effectors during colonization by mutualistic fungi is relatively unknown. This study identifies an extracellular ecto-nucleotidase and investigates the function of this effector in modulation of eATP in the apoplast and effects on immunity. The function of eATP has mainly been studied in pathogen infection. In this perspective the novelty of the work presented in the manuscript

is in extending the eATP biology to mutualistic fungi and in identifying a fungal effector ecto-nucleotidase.

Major concerns:

The eATP in Arabidopsis is estimated from the level in the surrounding medium while in barley the apoplastic fluid is isolated and the eATP measured. One would think that the level in the apoplastic fluid is a more precise methods and there is no explanation for the use of two different methods.

Response: We agree that measuring the levels of eATP in the apoplast is a more accurate analysis than in exudates. Nevertheless, the determination of metabolite concentrations in exudates is a generally accepted method and a good estimate of relative changes of metabolite pools in the apoplast. Unfortunately, it was not possible to extract apoplastic fluid from the roots of Arabidopsis seedlings because this tissue is very delicate. All our efforts to collect apoplastic fluids led to cytoplasmic contaminations (measured by LC-MS/MS proteomic approaches) in this host. We therefore decided to use a more robust approach even though less sensitive which is the measurement of the nucleotides in the surrounding media released by the plant roots. For barley the collection of apoplastic fluids from the roots of seedlings proved to be free from cytoplasmic contamination; thus we used this material for further analysis. We added this information in the new version: "It was not possible to extract apoplastic fluid from the roots of Arabidopsis seedlings for these experiments because this young tissue is very delicate and fragile. All our efforts to collect apoplastic fluids ended up with cytoplasmic contamination (measured by the LC-MS/MS proteomic approach) in this host" in the results.

This inconsistency is carried into the measurements of ecto-nucleotidase activity and eATP in the transgenic Ecto-5'NT lines. Here another method, membrane associated hydrolytic activity, is used while eATP is again measured in the medium. This is confusing and it would be more convincing and consistent to show results from apoplastic fluids, which is where the process is supposed and suggested to be localized.

Response: The E5'NT enzyme has a predicted membrane anchor which led to the decision to analyze its activity from membrane preparations. This method has two advantages.

- 1. This material can be purified making the measurement of activity reliable and not affected by possible enzymatic contamination from the cytoplasm and other apoplastic enzymes. Furthermore, this approach allows enzyme enrichment as well as the removal of nucleotides that are present in the apoplast.**
- 2. It demonstrates that the ectopic expression of the fungal E5'NT in plants leads to secretion of this enzyme and to a localization at the membrane, demonstrating the functionality of the SP and GPI anchor. This allowed the conclusion that the effects observed should be due to the activity of SiE5'NT as predicted during fungal colonization.**

In general, all different well-established methods for measuring the levels of eATP and the activity of an enzyme at the membrane led to consistent results. Therefore, the use of three different methods is not a drawback but a strong support for our conclusions.

One of the conclusions from this work is that immune signaling is influenced by the ecto-nucleotidase activity and expression of already known eATP response genes is measured to support this conclusion. However, immune responses in Arabidopsis are very well characterized and it is surprising that expression of central genes or even mutant normally used in immunity studies are not included in the manuscript.

Response: We used genes linked to ATP response as this was the focus of this work. These genes are reported in the paper by Choi et al., 2014 as marker genes for wounding and eATP signaling and accepted by the scientific community as marker genes. The selected genes are also responsive to *S. indica* colonization. We disagree with the reviewer in this case that more standard immunity markers should have been used especially when their connection to eATP is not clear.

The discussion of the results is not informative and results and conclusion obtained are not discussed in relation to current knowledge in plant-microbe interaction and immunity. Lines 335 - 393 is a review type summary and expression of the authors beliefs. Line 394 -423 just repeats observation

reported in the main text.

Response: We modified the discussion in the new version. We feel that we have deeply discussed the findings in light of the information available on ATP and nucleotides related immunity in planta and in animals and we have cited important publications. We are sorry if we missed some relevant literature here despite the careful searches we made. We also did not want to dilute the take home message by discussing immunity in a more general way.

Minor concerns:

The modelling of Pi-source effects, line 311- 333 is not useful without experimental support, as stands is appears like a speculative add-on.

Response: We do not agree that the modeling approach is just a speculative add-on. Instead, it points to the question “what is the fate of the released ATP?” and illustrates why it makes sense for the fungus to disintegrate the messenger molecule ATP and how to take advantage from the released phosphate also in the context of nutrition. It is rather straightforward to conclude that the cleaved phosphate groups from the ATP constitute an additional external phosphate source for both plant and fungus. Based on these conclusions the model illustrates the changes of the thermodynamic balance of nutrient fluxes from and to the plant, from and to the fungus and between them. From a thermodynamic point of view, it is clear that increased external Pi levels increase the driving force of Pi into the cells (or reduce the driving force out of the cells). We understand the scepticism about modeling and the conclusions drawn from them as this type of approach is not (yet) standard in biological sciences. However, in the context presented in the manuscript, additional experiments would just confirm the validity of basic thermodynamic principles, which are otherwise widely accepted. Additionally, we added this sentence in the results: “These new insights might guide future experiments to clarify whether the activity of SiE5’NT in the apoplast could serve both, nutritional needs and modulation of host immunity”. Models are rarely used in these kind of studies to support hypotheses and we think that modelers would appreciate it but we can remove the model if the editor insists.

Nomenclature used for constructs for example "Potef::mCherry, Potef::E5’NTwoGPI and Potef::E5’NTwoSPwoGPI" is clumsy and almost unreadable.

Response: Thank you for this helpful comment. We simplified the nomenclature as much as possible.

Referee #2:

In the present study, the authors report LC-MS/MS proteomics studies on the apoplastic fluid of barley roots following colonization with the beneficial fungus *S. indica*. Of the identified 102 fungal proteins, the authors put a particular focus on an ecto-5’-nucleotidase (E5’NT) among several ATP-scavenging enzymes that are specifically present in the fungal colonized roots. The authors further show that eATP levels are increased in response to *S. indica* colonization in barley and *Arabidopsis* roots, and that the eATP receptor DORN1 restricts fungal colonization. Based on sequence alignment and structure prediction, the authors suggest that the fungal E5’NT acts as a monomer and hydrolyzes different nucleotides. In ex vivo expression systems in *Arabidopsis* and in *Ustilago maydis*, the introduction of SiE5’NT exhibits a significant increase in the hydrolysis activity for ATP, ADP and AMP. Moreover, the authors show that transgenic expression of SiE5’NT in *Arabidopsis* results in enhanced colonization with *S. indica* and another pathogen, *C. incanum*, in line with a decrease in eATP levels. Based on these results, the authors propose a model in which plant-inhabiting fungi, including *S. indica*, secrete eATP-hydrolyzing enzymes to dampen an important branch of DAMP-mediated host immunity for promoting fungal colonization. Overall, this work has the potential to provide significant insight into beneficial plant-microbe associations and colonization strategies of plant growth-promoting and pathogenic fungi, important topics that attract high attention from the society. However, in my view, this work is incomplete in the following aspects.

1) There is not direct evidence that E5'NT-mediated decrease of eATP levels suppresses DORN-mediated defense responses. It remains not clear whether the observed degree of decrease in eATP levels is sufficient to attenuate DORN1 signaling or whether the eATP pools monitored by DORN1 are influenced by E5'NT in the transgenic plants or during fungal colonization. It would help if the authors test E5'NT effects on defense responses to exogenous eATP application.

Response: The experiment suggested by this reviewer is something we are currently doing in the scope of another project and only have preliminary data up to now. In general, the eATP responsive genes are not very highly upregulated and show a very quick response. Therefore, we aim to compare this data to data where we incubate ATP with recombinant E5'NT (production in progress) or buffer and measure defense responses in WT and *dorn1* mutant lines after application of ATP or E5'NT treated ATP (using the eATP marker genes described in the paper by Choi et al. 2014). This might help to clarify the response of the plant to eATP and its hydrolysis products.

2) As the authors mention in the text, changes in eATP levels can influence different pathways in plants. It is important to determine whether, and if so to which extent, the E5'NT-mediated increase of fungal colonization is DORN1 dependent, by testing E5'NT introduction (overexpression) effects in the *dorn1* mutant background. This will also help address the above concern (1).

Response: The proposed experiment is in principle a very good suggestion. If the SiE5'NT phenotype would solely be the result of lowering the eATP concentration, the expression of E5'NT in *dorn1* background should indeed result in the same *S. indica* colonization level as seen in *dorn1*. We tried to cross these lines but we have not yet succeeded. Such an approach will require about 2 years of work for producing and more importantly characterizing these lines, as they can only be tested in the third, homozygous generations and would require more than 3-4 biological repetitions of colonization experiments. Additionally, since both the E5'NT lines and the *dorn1* lines are already better colonized there are doubts that crossing these two lines will result in a measurable phenotype that is significantly different to the parental lines.

In Figure 4F we show that there are differences between E5'NT and *dorn1* in response to hyper-colonization by *S. indica*. The results suggest that the presence of E5'NT leads to better colonization but to a significantly lower induction of defense compared to the situation found in the *dorn1* mutant line. Therefore, we speculate that an additional (at the moment unknown) mechanism is involved. This could be, as discussed in the manuscript, the effect of an end product, e.g. adenosine which in animal systems was reported to have an immune-suppressive activity (*Staphylococcus aureus* synthesizes adenosine to escape host immune responses, <http://jem.rupress.org/content/206/11/2417.long>). From our metabolomics data from apoplastic fluid in barley we indeed could detect an accumulation of adenosine upon fungal colonization at 6 dpi (data not shown) but also other modified nucleotides accumulated which could be derived from synergistic activities of E5'NT and a second fungal derived enzyme identified from the LC-MS/MS analysis. At the moment we are following this line of research but this will take some time to be proven and goes beyond the scope of this paper. We added this information in the discussion.

3) What is the difference in *S. indica* colonization between Figs 4C and E? The results should be the same but very much different for the same lines used.

Response: The results are in line with each other. We used genomic DNA (gDNA) in one case (3, 5 and 7 dpi, 4C) and cDNA in the other case (5 dpi, 4E) due to the experimental setup. In the second case we wanted to measure also the expression of marker genes involved in eATP responses. Both methods are accepted measurements of fungal colonization. The normalization to plant DNA was done using AtUBI in 4C, a standard marker for plant gDNA levels used in our group and proved to be robust. AtSAND was used for normalization in 4E since this was the gene selected for the expression analysis by Choi et al., 2014. We wanted to be able to directly compare our results with those reported by Choi. Normalization using primers for AtUBI from the cDNA generated similar results. The scale is thus different between the graphics in figure 4C and 4E but not the results which show in both cases better colonization in the same range by the E5'NT expression line compared to the control lines at 5 dpi.

4) It also remains to be addressed whether and how increased *S. indica* colonization when eATP-triggered defenses are compromised influences different benefits of the fungal association. Are the previously described *S. indica* benefits reduced or increased in E5'NT-expressing transgenic plants and *dorn1* mutant plants? Induction of defense marker genes following *S. indica* inoculation was rather increased when eATP-triggered defenses were compromised, which makes me wonder what happens to the plant-fungus association afterward.

Response: The results while quite interesting would not give any insights on how the activity of E5'NT alters the plant immune response and therefore go beyond the scope of our paper. Unfortunately, the time points analyzed in this paper do not allow us to answer this question. The growth promoting effects are visible at later stages around 3 weeks after inoculation. This would require a completely new set of experiments.

5) In principle, the substrate specificity of the E5'NT protein needs to be tested in vitro by using recombinant protein purified from the cells. Otherwise, it is better to tone down discussions regarding the substrate specificity.

Response: We fully agree and we removed the claim about specificity.

6) To raise the hypothesis that eATP hydrolysis also contributes to nutrition, the authors should discuss possible co-expression of e.g. phosphate transporter genes with E5'NT.

Response: In general it is well accepted that phosphate transporters are an integral part of the transporter network involved in plant-fungus interaction and are tightly regulated. It is a very good suggestion to assess the expression of the Pi transporters in the E5'NT and *dorn1* lines and we will do this in the future. The setting used in the current approach would not allow us to see response at the transcript levels because the medium used was phosphate rich. We plan to test the transgenic lines under phosphate starvation and monitor fungus and plant responses to this situation.

Detailed comments.

Lines 84-87: The logical flow seems to be abrupt. Was there any link previously known between *S. indica* and eATP-mediated immunity? If so, please explain it.

Table S1 is missing (which could be my fault).

Figure legends should be more kind and self-explanatory. For instance, what are S1, S2d or S3d etc. in Fig 1B? When were the fungal inoculation data obtained (how many dpi)?

Response: S1 = Sample 1, S2d = sample 2 deglycosylated and so on. The dpi when the APF was harvested is indicated on top of the S number in figure 1B and in the respective supplementary table. We kept the nomenclature according to the original labelling of the MS data in order to always be able to match the information to the MSMS files. We added this info in the legend.

Referee #3:

In this comprehensive study Nizam et al. describe 1) a protein MS-based survey of the barley root apoplast during colonization with the beneficial and root-associated fungus *Serendipita indica*, indicating an enrichment of apoplastic fungal proteins related to purine metabolism 2) report a drastic increase in eATP levels at early stages of fungal host colonization in the barley and *A. thaliana* root apoplast as well as a hypercolonization phenotype of *A. thaliana dorn1* mutant roots known to be impaired in the perception of eATP. The authors then have 3) chosen one of the secreted potentially ATP-scavenging enzymes of *S. indica*, denoted PIIN_01005, for in-depth functional analysis. They demonstrate 4) evidence for ecto-5'-nucleotidase activity by heterologous expression of *S. indica* PIIN_01005 in the Basidiomycete *Ustilago maydis* and *A. thaliana* transgenic plants using washed culture suspensions and enriched plasma membrane fractions, respectively. 5) A *S. indica* hypercolonization phenotype is reported in *A. thaliana* transgenic lines constitutively expressing PIIN_01005 and this 6) correlates with a slight reduction (~10%) in eATP

levels at 5 days post inoculation (dpi). 7) An enhanced disease susceptibility phenotype is found in the *A. thaliana* transgenic line constitutively expressing PIIN_01005 upon inoculation with the root fungal pathogen *Colletotrichum incanum*. Finally, 8) metabolic modelling was applied to infer that eATP hydrolysis might serve a role in Pi nutrition for the fungus and in modulating plant innate immunity.

Very little is known about the functions of eATP during colonization with pathogenic and beneficial microorganisms. Thus, the present study has the potential to significantly advance our knowledge in this understudied research area in plant-microbe interactions. Overall the manuscript is well written and the Figures are organized in a logical sequence. The first two results sections describing the protein MS-based analysis of the root apoplast as well as *S. indica*-induced massive increases in eATP levels are largely solid and impressive (Fig. 1 and Fig. 2).

My major concern(s) with this work begins with the functional analysis of *S. indica* PIIN_01005, all inferences made from Fig. 5 plus the corresponding supplementary Figures as well as with conclusions based on the metabolic modelling (Fig. S10) as listed below. Unfortunately, the results corresponding to Fig. 5 are central for the major claim of this study and additional experimentation is critical to convince this reviewer. In addition, the Method section is in several cases fragmentary so that it is difficult to reconstruct how some of the experiments were conducted, how data were acquired and/or were processed. Below I have described my major points of critique.

1) GO-term enrichment analysis indicated an 'overrepresentation of apoplastic *S. indica* proteins implicated in purine/ATP metabolism' (Table S3, Fig. S1B and S2), suggesting the involvement of diverse extracellular proteins modulating eATP levels. I have difficulties to infer this conclusion from Fig. S2 given the lack of any detailed information in the corresponding Figure caption, Method section, and absence of applied statistical tests with corresponding p-values. Likewise, the authors state that *S. indica* PIIN_01005 is one of the 'most abundant' secreted fungal proteins (ln. 90 to 92). If this conclusion is based on protein MS data, is this supported based on both iBAQ values within samples and LFQ intensities across samples? Ideally, there should be correspondence between these two metrics, but the Method section remains entirely unclear on this point.

Response: We added in the new version a table with the raw data to improve readability of the GO term enrichment analysis (Table S3B-C). Please note that figure S1B is now figure S2 and S2 is now figure S3.

We agree with the reviewer that a normalization method could have been used. Indeed, we had several internal discussions on how to show these data in the most comprehensive way. Even though there could be different ways of presenting this data, we decided to use counts of uniquely matching fungal derived peptides, which is an accepted way. For us this seemed to be the best way of showing with confidence the presence of these proteins in the samples and still to keep a correlation to the real abundance of the fungal proteins considering that the same amount of plant material and of apoplastic fluid was submitted to LC-MC/MC analyses for each sample. We are aware that spectral counting could be a more accurate way but this is not going to change the conclusion that E5'NT is routinely found in the apoplastic fluid of barley at 3 different time points in the different replicates. We changed the wording in the text avoiding claims about abundance. We used in this version: "consistently found in the apoplastic fluid".

2) Key for the major claim of the present study are transgenic *A. thaliana* lines expressing *S. indica* PIIN_01005 encoding SiE5'NT. If my reading of the results section and Figures is correct, then a single *A. thaliana* over-expression line ('L3') was used for quantifying Ecto-5'NT activity and for inoculation experiments with beneficial *S. indica* and pathogenic *C. incanum*. If so, this is inadequate and evidence from further independent transgenic lines needs to be provided.

Response: In the previous supplementary figure S6, now S10 we show data from additional independent lines. The reviewer most likely has overseen this supplementary figure.

More importantly, the conclusions of the authors are in my view not justified because critical control transgenic plants expressing a catalytically inactive form of SiE5'NT are lacking. This is necessary to exclude the possibility that the modest reduction seen in eATP levels in the tested transgenic lines at 5 dpi of *S. indica* (Fig. 4D) as well as enhanced *S. indica* root colonization (Fig. 4C) is really the

result of ectopic SiE5'NT activity and not due to an unspecific perturbation of root-associated processes independent of SiE5'NT enzyme activity. The data shown with transgenic plants overexpressing mCHERRY are an insufficient control in this context (Fig. 4). Meaningful conclusions are further complicated by the fact that SiE5'NT enzyme activity appears to be compromised in control plants expressing a SiE5'NT-mCHERRY fusion protein (Fig. 4).

Response: This is a very good point. We are fully aware that the impact of E5'NT on the eATP level seems small, but a reduction of 10% should not be underestimated as it could affect the time by which the plant reaches a specific threshold for triggering a defense response. It should be considered that eATP levels are very tightly and actively regulated by the plant. We thus think that small effects on these levels and the timing affect colonization positively or negatively. In barley, for example, addition of 100 μ M ATP 24 hours before inoculation with *S. indica* led to a decreased fungal colonization in 4 independent biological experiments (data shown below):

A possible explanation to the effects observed during colonization of Arabidopsis transgenic lines could well be the accumulation in the apoplast of end products from the activity of the E5'NT, such as adenosine, a potent immune suppressor in animal systems, and/or phosphate, a regulator of compatibility in mycorrhizal associations. We comprehensively discuss this possibility in the paper. At this stage there is no evidence for an additional function independent of the hydrolytic activity. In this context speculation on an additional function would just side-track, but we mention this possibility in the discussion now: “Although our data indicate that the activity of the SiE5'NT is important for its function in the apoplast, we cannot fully exclude the possibility that an additional function independent of the hydrolytic activity is responsible for the immune suppressive effects observed in this study. Enzymatic inactivation by targeted point mutations in the catalytic domain could help address this possibility.”

More importantly, the manuscript does present control transgenic plants expressing a catalytically inactive form of SiE5'NT. We generated 3 types of transgenic Arabidopsis: (1) Plants expressing full-length SiE5'NT. In these plants we could demonstrate the catalytic activity of the enzyme.

(2) Plants expressing a SP_{E5'NT}:mCherry:E5'NT fusion, where the mCherry protein was integrated between the signaling peptide necessary for protein secretion and the rest of the SiE5'NT protein. This fusion protein can be detected (thanks to the mCherry tag) and was shown to be secreted to the apoplast. Thus, it appears structurally intact. For unknown reasons, the fusion protein did not exhibit catalytic function.

(3) Plants expressing cytosolic mCherry.

Plants of type (2) and (3) did not differ from each other in terms of eATP accumulation, fungal colonization and At1g58420 marker gene expression. Thus, there is no difference between plants expressing a catalytically inactive form of SiE5'NT and plants, which do not express any SiE5'NT protein.

The data presented shows that there is a clear correlation between activity and increased colonization.

Surely modification in the active site as suggested by this reviewer would be a more elegant control. Although biochemical information for fungal E5'NT are not available we could have tried to design a loss of function based on the information available from the human E5'NT but also there we would need to prove the activity and stability of these protein mutants in order to identify the best residue for mutation. Yet, while this is possible to do, the analysis would require a substantial amount of work and time and the outcome is uncertain. It is unclear why the active dead mCherry fusion protein variant in the respective Arabidopsis line is not a sufficient control to this reviewer for this first report.

A flaw in experimental design is the absence of epitope-tagged functional and catalytically inactive SiE5'NT variants expressed in transgenic *A. thaliana*, which would enable the authors to quantify steady-state levels of the plant-expressed SiE5'NT protein in the root apoplast and plasma membrane fractions and relate this to SiE5'NT enzyme activity as well as fungal infection phenotypes.

Response: Although this would help quantify steady-state levels of the protein it would most likely affect the activity of the native E5'NT as discussed above.

Thus, given the massive increases in eATP levels in the root apoplast at early stages of *S. indica* root colonization of wild-type plants (Fig. 2A and B), I am not convinced that the modest decrease in eATP levels seen upon SiE5'NT overexpression in transgenic plants (Fig. 4D) is causally and directly linked to the observed changes in *S. indica* and *C. incanum* infection phenotypes.

Response: We are of the opinion that it is central to point out the importance of bioactive extracellular nucleotides in plant fungus interactions and to open this field of research. We are aware that we did not fully clarify the mechanism/s behind the observed effects but it is of paramount importance to show that the activity of an extracellular fungal enzyme mediates compatibility (in beneficial as well as detrimental fungal plant interactions) and we believe that our data in this MS allow this conclusion. We largely rephrased the discussion and results in order to address the limitations of our study.

3) Conclusive evidence for SiE5'NT physiological functions also demands the generation of corresponding *S. indica* knock-out mutants and subsequent inoculation experiments with the mutant strains. I understand this is at present technically challenging with *S. indica* and might not be realistic, but the lack of this genetic tool further limits an unambiguous assessment of SiE5'NT function(s) during plant-fungus interactions.

Response: Currently generating a knock out is not possible in *S. indica*. This fungus is dikaryotic and thus possesses two haploid nuclei (heterokaryotic). Simultaneous knock out of the two copies of a gene is thus quite inefficient. In addition, heterokaryotic fungi also have relatively lower rates of homologous recombination, further making the gene knock out process inefficient in this fungus. We are establishing Cas9 and we hope in future to solve this issue.

4) Although I am appreciative of metabolic modelling, I have not only difficulties to reconstruct from the method section the calculation of the sparse dataset visualized in Fig. S10, I also believe the conclusion drawn remains speculative unless underpinned by advanced stable isotope labelling experiments. I suggest removing this part from the manuscript as these are premature observations that are not central for the main claim of the work.

Response: Concerning the comment on the methods section, it should be noted that the modeling approach was described in detail in the original publication (Schott et al., 2016). Unfortunately due to space limitation this cannot be repeated in the current manuscript. Concerning the conclusions drawn: The model points to the question "what is the fate of the released ATP?" and illustrates that it makes sense for the fungus to disintegrate the messenger molecule ATP and to take advantage from the released phosphate also in the context of nutrition. As already written above, it is rather straightforward to conclude that the cleaved

Pi's from the ATP constitute an additional external phosphate source for both plant and fungus. Based on these conclusions the model just illustrates the changes of the thermodynamic balance of nutrient fluxes from and to the plant, from and to the fungus and between them. From a thermodynamic point of view, it is clear that increased external Pi increases the driving force of Pi into the cells (or reduces the driving force out of the cells).

Minor point:

Ln. 195: replace 'hypnotized' by 'hypothesized'

Response: We modified this.

1st Revision - authors' response

5 December 2018

Response to the Senior Editor

As far as I understand, the manuscript does not provide direct data that show an effect on the plant immune response. This was also mentioned by the referees. The response shown is to eATP. I therefore think that the statements on the plant immune response in the title, abstract and discussion (when referring to Fig 5) need to be toned down.

We agree with the Editor and have toned down our statements and changed the wording in the respective passages (visible by track changes). We also changed the title. Please note that figure 5 is now figure 6.

The manuscript has 5 main figures but separate results and discussion sections. Either one more main figure needs to be added/moved from the supplementary figures, or the results and discussion sections need to be combined. This is the difference between our scientific reports and full articles. You can find more information in our guide to authors online.

We provide now a new main figure (Fig. 5) showing the thermodynamic model. This figure was present in the Expanded View section in the last version of the manuscript (previous Figure S 10).

Fig 2A does not specify the p-value and tests, Fig 4E does not specify "n" and the error bars, SF6 does not specify the bars and error bars, and SF8C mentions n=2 in which case no error bars can be shown. Fig 8E needs to explain the box plots.

We have added the information to the figure legend of Fig 2A reading now 'Error bars show the standard error of the mean obtained from three independent biological replicates at $p < 0.05$ (*), 0.01 (**) analyzed by Student's t-test.'. The additional information to figure 4E are as follow: 'Error bars show the standard error of the mean obtained from three independent biological replicates at $p < 0.05$ (*), 0.01 (**) analyzed by Student's t-test.'

Supplementary figure S8C now figure S9C: Error bars represent standard deviation of the mean from in total 12 plants of two independent biological experiments. We added this information to the figure legend.

Supplementary figure S8E now figure S9E: We explained the box plots. 'Box plots show the quartile of seed weights of the plant individuals analyzed. The overlaid dot plots shows the respective individual values.'

Tables S1-S5 should either be EV tables or Datasets (named Table EV1, 2, or Dataset EV1, 2, etc). All tables need titles and can have legends. All tables need to be uploaded as individual files, either word or excel. The Appendix table 1 also needs a title and may be legend, and it can also be either an EV table or a Dataset. A Dataset and EV table can have more tabs, but a regular table cannot.

We have changed this and provide separate excel files named Table EV1-5. We also added the table titles at the end of the main text. The mass spectrometric data file is now Dataset EV1 and a respective title is given at the end of the MS file.

The Appendix file needs to be a single file with table of content and the figure legends included. The nomenclature is Appendix Figure S1, Appendix Table S1, etc.

We removed completely the appendix and instead added the data as Dataset EV1.

The methods in the Appendix file should only stay there if they exclusively refer to data shown in the Appendix file. Otherwise they should be moved to the main methods.

All methods have been added to the material and methods section in the main text.

The text in the figures is often too small to be read. All text needs to be readable. You can check our figure guidelines here:

http://www.embopress.org/sites/default/files/EMBOPress_Figure_Guidelines_061115.pdf

We have reformatted our main figures to match the information given in the Figure guidelines.

You mention "data not shown" on page 10 and 13, which we don't allow. Please either remove the statement or may be use "personal communication".

We have changed this in the manuscript.

Section B in the Checklist on statistics needs to be completed.

We have completed the statistics description in the Checklist. In addition, all statistical analysis methods, number of replicates and error bar descriptions are given in the manuscript file (either figure legend or Material and method section). We hope that we have spotted everything we missed earlier and added all the respective information.

EMBO press papers are accompanied online by A) a short (1-2 sentences) summary of the findings and their significance, B) 2-3 bullet points highlighting key results and C) a synopsis image that is 550x200-400 pixels large (the height is variable)

We have given the short (1-2 sentences) summary, 2-3 bullet points highlighting key results in the main text before the abstract and we have uploaded the synopsis image

2nd Editorial Decision

11 December 2018

Thank you for the submission of your revised manuscript. I am afraid that a few more changes are still needed.

I changed two words in the abstract to tone down the conclusions. Please let me know if you agree. I also modified the short summary and bullet points, please have a look at the attached manuscript word file with tracked changes and let me know what you think.

The Appendix needs to be called Appendix (with the supplementary figures called Appendix Figure S1, etc) and the figure legends need to be included in the Appendix file. The Appendix file also needs a table of content. Please make sure that the callouts of the Appendix figures in the manuscript text are correct.

The Dataset in the Appendix needs to be taken out and uploaded as separate word or excel file called Dataset EV1 or EV2, as you already have another dataset called EV1. All Datasets need titles in their files, eg in the first tab of an excel file.

It is unclear to me whether you have submitted proteomics data to a public database. If this is the case, please add a "Data Availability" section to the materials and methods and list the accession codes.

The table and Dataset titles need to be removed from the main manuscript text and added to the table and dataset excel or word files.

You can bring forward all manuscript files to a new version (V3) of your manuscript and then replace only the files that need to be replaced.

The authors performed all minor editorial changes.

Corresponding Author Name: Alga Zuccaro

Manuscript Number: EMBOR-2018-47430